# ConText: Driving In-context Learning for Text Removal and Segmentation

**Fei Zhang** [1 2 *]  **Pei Zhang** [3]  **Baosong Yang** [3]  **Fei Huang** [3]  **Yanfeng Wang** [1]  **Ya Zhang** [1 †]

## Abstract

This paper presents the first study on adapting the *visual in-context learning* (V-ICL) paradigm to optical character recognition tasks, specifically focusing on text removal and segmentation. Most existing V-ICL generalists employ a reasoning-as-reconstruction approach: they turn to using a straightforward `image-label` compositor as the prompt and query input, and then masking the query label to generate the desired output. This direct prompt confines the model to a challenging single-step reasoning process. To address this, we propose a *task-chaining* compositor in the form of `image-removal-segmentation`, providing an enhanced prompt that elicits reasoning with enriched intermediates. Additionally, we introduce *context-aware aggregation*, integrating the chained prompt pattern into the latent query representation, thereby strengthening the model's in-context reasoning. We also consider the issue of visual heterogeneity, which complicates the selection of homogeneous demonstrations in text recognition. Accordingly, this is effectively addressed through a simple *self-prompting* strategy, preventing the model's in-context learnability from devolving into specialist-like, context-free inference. Collectively, these insights culminate in our **ConText** model, which achieves new *state-of-the-art* across both in- and out-of-domain benchmarks. The code is available at https://github.com/Ferenas/ConText.

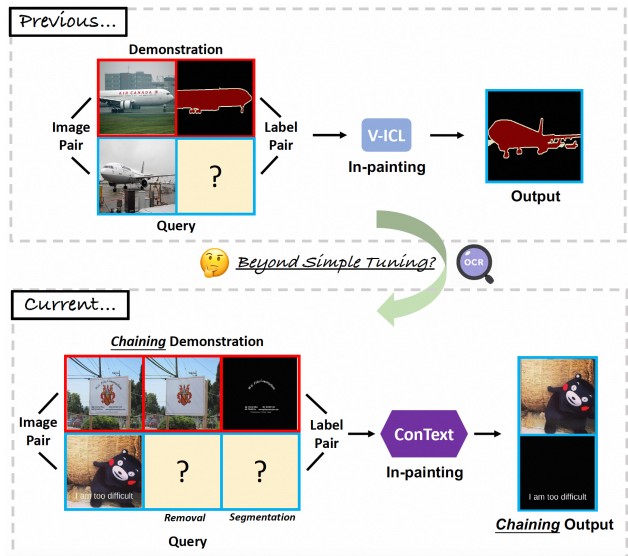

*Figure 1.* Comparison with previous V-ICL paradigm and our proposed OCR-adapted in-context system. Instead of relying simply on task-specific tuning, our **ConText** focuses on chaining together related visual tasks to leverage their mutual benefits, thereby embracing a more powerful in-context understanding and reasoning.

## 1. Introduction

Recent years have witnessed significant advances due to the emergence of *large language models* (LLMs) (Brown et al., 2020; Touvron et al., 2023), enabling models with powerful reasoning capabilities. Notably, *in-context learning* (ICL) (Rubin et al., 2021; Dong et al., 2022; Wies et al., 2023), a method deserving of particular attention, empowers models with training-free learning ability by using merely a few input-output examples (demonstrations) as the context. This efficient LLM-inspired paradigm has sparked interest among computer vision researchers, leading to the exploration of *visual in-context learning* (V-ICL) paradigm (Bar et al., 2022; Wang et al., 2023b;d; Bai et al., 2024), and paving the way for developing vision-centric contexts.

Similar to ICL defined in natural language processing, current works have regulated V-ICL as a reasoning-from-demonstration process, where an in-context prompt is defined as an `image-label` pair, guiding any query image input to generate its target output. To instantiate this, mainstream approaches (Bar et al., 2022; Wang et al., 2023b;d; Fang et al., 2023; Wang et al., 2024) proposed to composite two `image-label` pairs into a single visually grid-like input, with one pair serving as the query by masking the label regions, and then perform mask-wise reconstruction based on MAE (He et al., 2022). This in-painting-based baseline

*Work was done during the internship at Tongyi Lab <ferenas@sjtu.edu.cn>. [1]Shanghai Jiao Tong University (Yanfeng Wang and Ya Zhang are with School of Artificial Intelligence) [2]Shanghai Innovation Institute [3]Tongyi Lab, Alibaba Group. Correspondence to: Ya Zhang <ya_zhang@sjtu.edu.cn>.

*Proceedings of the 42nd International Conference on Machine Learning*, Vancouver, Canada. PMLR 267, 2025. Copyright 2025 by the author(s).

nurtures an effective in-context model by implicitly learning the `image-label` mapping from the given context.

In contrast to the above studies focused on natural-object-oriented generalists, this paper introduces the first V-ICL framework tailored for *optical character recognition* (OCR) tasks, including text segmentation and removal. Intuitively, the straightforward approach goes to task-specific fine-tuning of existing MAE-based pipelines, as seen in Wang et al. (2023d); Fang et al. (2023); Wang et al. (2024). While effective, this method often leads to a *single-task-centered* paradigm, limiting the model to straightforward `input-output` mappings and single-step reasoning. This raises an intriguing question: *Is immediate reasoning the most suitable approach for visual tasks?* To enhance the reasoning abilities of LLMs, Wei et al. (2022); Wang & Zhou (2024) have proposed to exploit the task-wise correlation that chaining relevant task as one holistic prompt, yielding a comprehensive reasoning with enriched multi-task information. Inspired by this, our paper explores linking multiple visual tasks to create a *task-chaining* in-context prompt. This approach explicitly enhances generalized reasoning capabilities through the integration of multi-task rationales, thereby facilitating more powerful ICL inference.

To this end, we propose **ConText**, an enhanced V-ICL framework specifically designed for text removal and segmentation tasks. Considering the implicit logic shared between these tasks, we propose restructuring the original single-reasoning prompt (`input-output`) into a *task-chaining* format, `input-rem-seg`, where the masking reconstruction process is applied to both task labels. This restructuring transforms our in-context generation into an end-to-end multi-task generalist. Building on the insights of Wang et al. (2023f); Yu & Ananiadou (2024), who revealed that the query label plays a crucial role in ICL reasoning by consolidating all demonstration-level information, we aim to enhance the model's reasoning-by-demonstration capabilities. To this end, we design the *context-aware aggregation* (CAA) module that explicitly integrates prompt knowledge patterns into the query feature, thereby improving the model's contextual understanding. Additionally, given the inherent heterogeneity in text recognition, finding an appropriate "same-class" in-context prompt for the query input poses a challenge. To address this, we propose a simple yet effective training technique named the *self-prompting* strategy, which periodically uses the same visual demonstration as the input query. Experimentally, this strategy significantly aids in maintaining generalized in-context reasoning, preventing the model from devolving into a specialist that reasons without demonstration. In summary, our overall contributions are as follows:

- We propose the *task-chaining* prompting that enables visual in-context reasoning with explicit task-wise inter-

mediates. In this way, this enriched demonstration is encouraged to comprehensively improve the model's ICL capabilities with exploiting multi-task logic.

- We propose **ConText**, the first OCR-focused V-ICL generalist that enhances features through explicit *context-aware aggregation*, and ensures text-level in-context learnability via self-query recognition, leading to significant advancements in generalized in-context reasoning.

- Extensive results on several benchmarks demonstrate the general superiority and effectiveness of our method compared to all baseline V-ICL generalists and specialists, yielding new *state-of-the-art* (SOTA) performance on both text removal (**+4.50** PNSR) and segmentation (**+3.34%** fgIoU). Surprisingly, **ConText** also emerges amazing *training-free reasoning* prompted from human-oriented visual instructions, sufficiently exhibiting its comprehensive in-context inference abilities.

## 2. Related Work

### 2.1. In-context Generalists

**Mechanistic Exploration.** To uncover the mystery of ICL, many studies have extensively explored the mechanistic interpretability. Theoretical approaches formalized ICL either as an implicit functional learner using standard algorithms (Xie et al., 2021b; Garg et al., 2022; Akyürek et al., 2022; Li et al., 2023c), or as a meta-learner performing internal gradient descent based on demonstrations (Dai et al., 2022; Von Oswald et al., 2023). Empirical studies (Min et al., 2022; Su et al., 2022; Mavromatis et al., 2023; Li & Qiu, 2023; Pan, 2023; Liu et al., 2023) have examined latent feature changes with demonstration-level operations like replacement, reformatting, and ordering. Building on these insights, Wang et al. (2023a); Yu & Ananiadou (2024) have concluded that label words play a crucial role in extracting and synthesizing the input information within the demonstration during ICL inference. Inspired by this input-label interplay, this paper introduces an innovative context aggregation module to explicitly enhance the visual representation of labels, thereby augmenting the reasoning abilities of ICL.

**V-ICL.** The advancement of *visual in-context learning* (V-ICL) in computer vision has been slow due to diverse and complex task types. Early attempts focused on effective V-ICL representation, with Bar et al. (2022) pioneering a composited-prompting pattern using MAE (He et al., 2022) to perform mask-targeted in-painting on images created by concatenating one task-specific input-output pair with a query-mask image pair. To enhance its in-domain performance, Zhang et al. (2023b); Sun et al. (2023); Zhang et al. (2024b) have focused on visual retrieval for optimal demonstration selection. Building on Bar et al. (2022), some studies have used additional curated data to train/fine-tune MAE-like models, improving ICL across tasks (Wang et al.,

2023b) or specific domains like segmentation (Wang et al., 2023d), skeleton recognition (Wang et al., 2024), and 3D point cloud analysis (Fang et al., 2023). Beyond this MAE-based implementations, Wang et al. (2023f) enhanced the stable diffusion (Rombach et al., 2022) with ICL via conditional fine-tuning, while Bai et al. (2024) explored sequential modeling for visual auto-regressive generation. This paper proposes an OCR-targeted ICL minimalist based on a MAE-like architecture, which enables concurrent multi-task inference by leveraging task-wise correlations.

## 2.2. OCR-targeted Specialists

**Scene Text Segmentation.** Scene text segmentation focuses on pixel-level character recognition, a derivative of foreground-background segmentation tasks. Initially, traditional methods like thresholding (Otsu et al., 1975) and low-level features (Vo et al., 2018) struggled with complex colors and textures. Recent advances have seen deep learning methods like SMANet (Bonechi et al., 2019), ARM-Net (Ren et al., 2022), and TextFormer (Wang et al., 2023c), which incorporated multi-scale attention, high-level semantics, and enhanced text detail perception, respectively. Additionally, character/line-level discriminators (Xu et al., 2021; 2022) have been utilized. The advent of *vision transformers* (ViT) (Dosovitskiy et al., 2020) has led to efficient fine-grained text segmentation approaches (Yu et al., 2023b; 2024; Ye et al., 2024), with Hi-SAM (Ye et al., 2024) employing SAM (Kirillov et al., 2023) for a generalized framework. Unlike these discriminative models, this paper proposes a universally generative-based framework for the task.

**Text Removal.** Text removal seeks to seamlessly replace text with coherent backgrounds. Early one-stage apples approaches combined text localization and in-painting within a single network using image-to-image translation techniques (Mirza, 2014; Phillip et al., 2017), but often left noticeable text remnants due to limitations in text perception. To improve precision, two-stage methods have gained traction by incorporating explicit text segmentation modules (Bian et al., 2022; Hou et al., 2022; Du et al., 2023a;b; Lyu et al., 2023) or using external text detectors (Zdenek & Nakayama, 2020; Tursun et al., 2020; Tang et al., 2021; Conrad & Chen, 2021; Liu et al., 2022b) to enhance text localization. Additionally, strategies such as coarse-to-fine (Liu et al., 2020; Tursun et al., 2020; Jiang et al., 2022) and multi-step progressive refinements (Lyu & Zhu, 2022; Wang et al., 2023e) have been explored for more comprehensive text removal. Despite the complexity of two-stage methods, ViT-Eraser (Peng et al., 2024a) showed that a streamlined one-stage framework using ViT can outperform these methods, offering a promising alternative. In this paper, we employ this method to erase images for segmentation benchmarks lacking human-annotated removal labels.

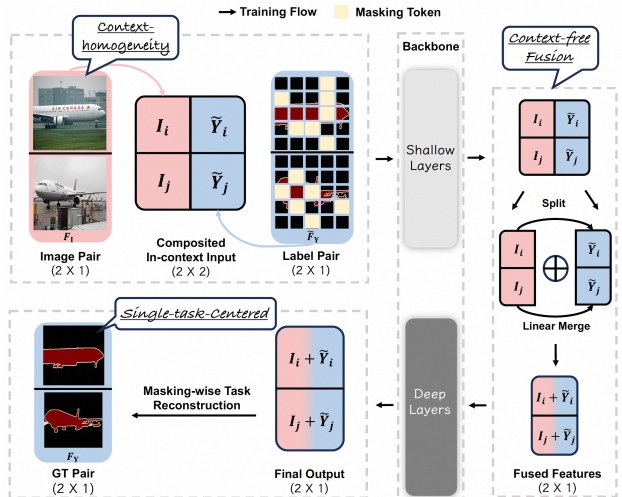

*Figure 2.* The *training* pipeline of previous V-ICL generalists (object segmentation as the illustrative task). This MAE-based framework formalize ICL as an composited-image (image-label) reconstruction process. During training, the two image-label pairs serve as the mutual in-context demonstration for each other, while during inference, only one pairs' label is masked to generate the query output. This baseline possesses 3 key characteristics integral to the foundation of establishing an OCR-targeted ICL paradigm.

## 3. Composited-Prompting V-ICL Generalists

As previously discussed above, most V-ICL paradigms artfully model the in-context inference as a composited image in-painting process based on MAE (He et al., 2022). Formally, given an image-label pair representing the input-output $(\mathbf{I}_i, \mathbf{Y}_i) \in \mathbb{R}^{3 \times h \times w}, i \in \{1, ..., n\}$, where $h \times w$ denotes the image size, and $n$ is the number of data samples. The final in-context visual input is generated by concatenating two image-label pairs at $h$-dimension, i.e.,

$$\mathbf{F} = [\mathbf{F}_\mathrm{I}, \mathbf{F}_\mathrm{Y}] = \begin{bmatrix} \mathbf{I}_i & \mathbf{Y}_i \\ \mathbf{I}_j & \mathbf{Y}_j \end{bmatrix} \in \mathbb{R}^{3 \times 2h \times 2w}.$$ During the training stage, certain label areas in $\mathbf{F}$ are erased, yielding $\widetilde{\mathbf{F}} = [\mathbf{F}_\mathrm{I}, \widetilde{\mathbf{F}}_\mathrm{Y}] = [\mathbf{F}_\mathrm{I}, \mathbf{M}_\mathrm{Y}\mathbf{F}_\mathrm{Y}]$. Here $\mathbf{M}_\mathrm{Y} \in \{0, 1\}^{3 \times 2h \times w}$ refers to the randomly generated mask, where 0 indicates the masked areas equipped with learnable masking tokens. With this global erasing operation, these two input-output pairs, without special distinction in demonstration-query, mutually serve as the in-context information to support each other in reconstructing the label targets. Specifically, after forwarding $\widetilde{\mathbf{F}}$ through an encoder-decoder backbone, the final output is used to predict the pixel values of the originally erased labels using the *mean squared error* (MSE) loss function. During the inference, masking tokens are applied merely to the label position of one of the input-output pairs ($\mathbf{Y}_i$ or $\mathbf{Y}_j$). Furthermore, to take advantage of the input image pair $\mathbf{F}_\mathrm{I}$, Wang et al. (2023b;d) proposed a feature fusion operation that integrates the features of the input image and output label at a shallow layer. This intuitive fusion not only brings a twofold decrease regarding the memory

costs, but also enhances the in-context representation of the label by using the input-to-output correspondence. Figure 2 presents the pipeline of these composited-prompting V-ICL frameworks, which shares 3 essential characteristics:

❶ *Single-task-centered.* This pipeline supports in-context inference for only one task at a time. As a result, evaluating multiple tasks necessitates multiple rounds of inference because of the straightforward image-label composition for constructing demonstration. Consequently, this single-task-centered mechanism is unable to leverage inter-task correspondence to enhance the generalized ICL capability.

❷ *Context-free fusion.* The input-output fusion operation, which involves a linear addition for each image-label pair, solely integrates information within each input-output pair itself. As a result, the combined labels primarily focus on extracting the visual patterns of individual input-output pairs, lacking the learnability from the other given context.

❸ *Context-homogeneity.* The two composited pairs shall remain the same objectness. For instance, segmenting an *airplane* must be instructed with another *airplane* image-mask demonstration. This visual object homogeneity, as it is easily defined and enriched, explicitly provides a diverse contextual environment, thereby leading to demonstration-sensitive V-ICL learnability (Zhang et al., 2023b; 2024a).

This paper aims to leverage the above pipeline to develop the first V-ICL model tackling two representative OCR tasks: *text segmentation and removal*. Beyond the intuitive task-specific fine-tuning as seen in Pan (2023); Wang et al. (2024), we are driven to explore targeted improvements on these inherent traits to enhance in-context performance.

## 4. Method

Figure 3 presents the overall framework of our proposed **ConText**, which integrates three specifically designed modules to improve the MAE-based baseline. The following part will provide a detailed explanation of each module.

### 4.1. *Task Chaining*: Beyond Single-task-prompting

As intuitively observed, there is an implicit inter-task logical connection between text segmentation and text removal: theoretically, *the segmentation mask should correspond to the visual difference between the original image and its erased counterpart.* Therefore, exploiting the task-level correlation shall bring expected advancement compared to single-task-prompting mechanism. To this end, we propose to recast the task demonstration by forming an explicit chain rather than a simple input-output pair. Specifically, we denote $\mathbf{O} \in \mathbb{R}^{3 \times h \times w}$ as the erased image, and we define a new prompt demonstration as $\mathbf{F} = [\mathbf{F}_\mathrm{I}, \mathbf{F}_\mathrm{O}, \mathbf{F}_\mathrm{Y}] = \begin{bmatrix} \mathbf{I}_i & \mathbf{O}_i & \mathbf{Y}_i \\ \mathbf{I}_j & \mathbf{O}_j & \mathbf{Y}_j \end{bmatrix} \in \mathbb{R}^{3 \times 2h \times 3w}$, where $\mathbf{Y}$

here denotes the segmentation mask. The observed improvements in a pilot experiment (please refer to Appendix A.1 for more details) verify the benefit of this task-level prompting. To further exploit this advantage, we implement the mask-then-reconstruct process on $\mathbf{F}_\mathrm{O}$ during the training, yielding $\widetilde{\mathbf{F}} = [\mathbf{F}_\mathrm{I}, \widetilde{\mathbf{F}}_\mathrm{O}, \widetilde{\mathbf{F}}_\mathrm{Y}] = [\mathbf{F}_\mathrm{I}, \mathbf{M}_\mathrm{O}\mathbf{F}_\mathrm{O}, \mathbf{M}_\mathrm{Y}\mathbf{F}_\mathrm{Y}]$. With maintaining the logical task-level connection, we set the mask as $\mathbf{M}_\mathrm{O} = \mathbf{M}_\mathrm{Y}$ by preserving their spatial correlations. Correspondingly, we turn to a weight-shared decoder for reconstructing each task using their corresponding labels, accompanied by different weight regularization. Reasonably, we also set the masking token to the removed query label position ($\mathbf{O}_i$ or $\mathbf{O}_j$) during the inference, thereby generating all task outputs in an end-to-end manner.

### 4.2. *Context-aware Aggregation*: Fusing-with-prompt

The findings (Wang et al., 2023a; Yu & Ananiadou, 2024) achieve a critical hypothetical consensus concerning the working mechanism of ICL: *the label position acts as the core for progressively extracting prior demonstration information in the shallow layers, with the final label absorbing all information* (refer to Appendix A.2 for a detailed explanation). Clearly, this label-anchor perspective partially supports the feasibility of the baseline early-fusion in V-ICL generalists, where the final output represents the task label by semantically integrating inner-demonstration knowledge. However, simply merging the individual input-output pair features shall weaken the label representation because of the absence of outer-demonstration fusion, yielding an insufficient understanding of the contextual prompt. To address this, we propose the *context-aware aggregation* (CAA) to strengthen the label in-context representation. Specifically, our fusion process 2 steps to form the final labels, and 1) goes to a similar *context-free fusion* that yielding $\widetilde{\mathbf{F}}_1 =$

$$[\widetilde{\mathbf{F}}_{\mathrm{O}1}, \widetilde{\mathbf{F}}_{\mathrm{Y}1}] = \begin{bmatrix} \mathbf{I}_i + \widetilde{\mathbf{O}}_i + \alpha_\mathrm{y}\widetilde{\mathbf{Y}}_i & \mathbf{I}_i + \alpha_\mathrm{o}\widetilde{\mathbf{O}}_i + \widetilde{\mathbf{Y}}_i \\ \mathbf{I}_j + \widetilde{\mathbf{O}}_j + \alpha_\mathrm{y}\widetilde{\mathbf{Y}}_j & \mathbf{I}_j + \alpha_\mathrm{o}\widetilde{\mathbf{O}}_j + \widetilde{\mathbf{Y}}_j \end{bmatrix} \in$$

$\mathbb{R}^{3 \times 2h \times 2w}$, where $\alpha_\mathrm{o}$ ($\alpha_\mathrm{y}$) is a learnable weight that regulates the chaining prompt from the removal (segmentation) counterpart (note that here we use the same notation to represent the latent feature for convenience). Intuitively, $\widetilde{\mathbf{F}}_1$ centers on the inter-demonstration fusion. To empower demonstration-aware context fusion, we propose 2) an additional cross-attention-based module CAA as $\widetilde{\mathbf{F}}_2 =$

$$[\widetilde{\mathbf{F}}_{\mathrm{O}2}, \widetilde{\mathbf{F}}_{\mathrm{Y}2}] = \begin{bmatrix} \phi(\widetilde{\mathbf{F}}_\mathrm{O}^i, \widetilde{\mathbf{F}}_j) & \phi(\widetilde{\mathbf{F}}_\mathrm{Y}^i, \widetilde{\mathbf{F}}_j) \\ \phi(\widetilde{\mathbf{F}}_\mathrm{O}^j, \widetilde{\mathbf{F}}_i) & \phi(\widetilde{\mathbf{F}}_\mathrm{Y}^j, \widetilde{\mathbf{F}}_i) \end{bmatrix} \in \mathbb{R}^{3 \times 2h \times 2w},$$

where $\phi(\texttt{query}, \texttt{key/value}) : \mathbb{R}^{3 \times h \times w} \rightarrow \mathbb{R}^{3 \times h \times w}$ denotes a shared-cross-attention mapping to the query feature. Based on a further combination as $\widetilde{\mathbf{F}}_1 + \widetilde{\mathbf{F}}_2$, each context-free label could be explicitly enhanced to extract the information from other demonstrations, resulting in a more comprehensive context understanding.

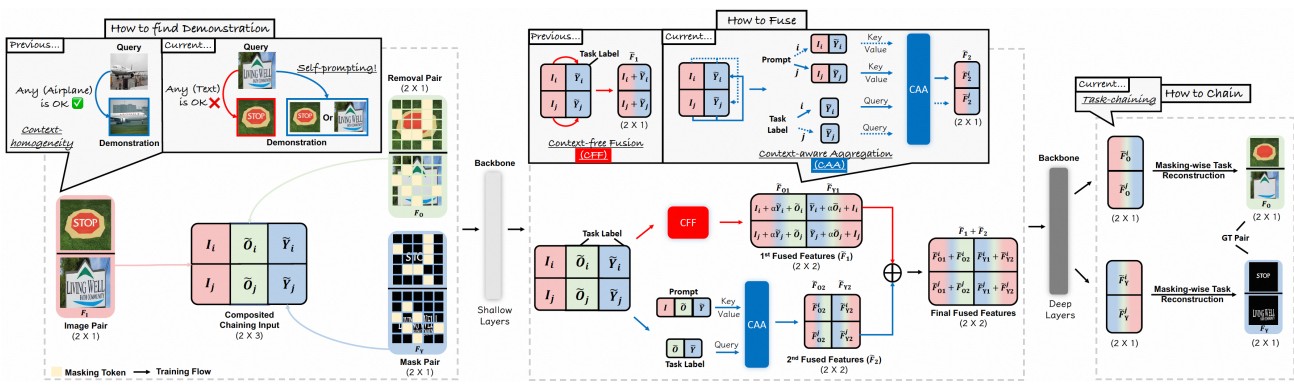

*Figure 3.* The *training* pipeline of **ConText**, a V-ICL framework for text removal and segmentation, enhances the baseline by leveraging inherent characteristics. We create an end-to-end multi-task generation through *task-chaining*. Additionally, our *context-aware aggregation* (CAA) enhances label representation towards better in-context understanding. We also employ a *self-prompting* strategy to ensure in-context learnability for text recognition. During inference, masking tokens are merely used for query removal and segmentation labels.

### 4.3. *Self-prompting*: Servicing In-context Learnability

In scene text recognition, the inherent heterogeneity of text is more complex compared to natural scene object recognition. This complexity arises from the diversity in fonts, styles, languages, and contexts, posing significant challenges for achieving homogeneous in-context demonstrations (Karaoglu et al., 2012). Consequently, employing a baseline training strategy with randomly selected examples may cause a generalist V-ICL model to degrade into a task-specific one. This was experimentally validated in Section 5.3, as only minor performance differences were observed between ground-truth-based and random demonstration. To address this, Souibgui et al. (2021); Sahay & Coustaty (2023) have introduced the "few-shot" learning concept in OCR tasks, suggesting the use of fragments of the query image itself as effective demonstrations. Inspired by this, we propose constructing model inputs by using two identical input-output pairs ($\widetilde{\mathbf{F}}_i = \widetilde{\mathbf{F}}_j$) with a certain probability. This approach is expected to enable the ICL model to maintain both task-specific reasoning and generalized in-context learnability. While this self-prompting strategy may seem simple, we emphasize that it is a crucial training technique for preserving the text-targeted ICL ability.

## 5. Experiments

### 5.1. Experimental Settings

**Tasks & Benchmarks & Evaluation Metrics.** Our work centers on two representative pixel-level OCR tasks, *text segmentation* and *text removal*. For text segmentation, we, following the majority of the pipelines (Yu et al., 2023a; Wang et al., 2023c; Yu et al., 2024; Ye et al., 2024), adopt four datasets with high-quality pixel-level labels: HierText (Long et al., 2022), TotalText (Ch'ng & Chan, 2017), ICDAR13 FST (Karatzas et al., 2013), and TextSeg (Xu et al., 2021). We use the *foreground Intersection-over-Union* (fgIoU) and F-score for evaluating the segmentation. For text removal,

we follow the prevailing pipelines (Du et al., 2023b; Peng et al., 2024a) and adopt two datasets: SCUT-EnsText (Liu et al., 2020), and SCUT-Syn (Zhang et al., 2019), where the latter one is a group of artificially synthesized data. Additionally, we incorporate HierText as another benchmark for this task by using the annotation from (Zhu et al., 2024). To evaluate the performance of removal, we use seven commonly-used image generation metrics: PSNR, MSSIM, MSE, AGE, pEPs, pCEPs, and FID. Note that fgIoU, MSSIM and MSE are presented in (%) in this paper.

**Implementation Details.** To train generalist, we have two different pipelines based on the training data volume: *i)* **ConText** with HierText *train* set & *ii)* **ConTextV** with (HierText + TextSeg + TotalText + SCUT-EnsText) *train* set. Here *i)* works for conducting the ablations of the designed modules, and *ii)* serves as task-specific comparison with the prevailing specialists. We use AdamW optimizer (Kingma & Ba, 2015) and a cosine learning rate scheduler, accompanied with a base learning rate of 0.0001, and weight decay of 0.1. The training epoch is set to 150, and the batch size is set to 2 with a two-step gradient accumulation. We adopt 16 A100 (80GB memory) to implement the training procedure, leading to a total batch size of 64. As the choice of visual demonstration shall have a considerable impact during the in-context inference (Rubin et al., 2021; Zhang et al., 2023b), we report the model's performance averaged among 3-times trial. More details could refer to Appendix B.1.

### 5.2. Global Comparison

**Comparison with ICL Generalists.** Our first experiment involves a general comparison with prevailing V-ICL frameworks, which serve as reasonable baselines. Table 1 provides a detailed comparison for text removal and segmentation. Note that all fine-tuning-based methods are trained solely on HierText and directly evaluated on other downstream datasets, which also demonstrates a model's out-

*Table 1.* Comparison with different V-ICL frameworks against several text removal (*Rem.*) and text segmentation (*Seg.*) benchmarks.

| Method | Text Removal (PSNR↑ / FID↓) | | | △ | Text Segmentation (fgIoU ↑) | | | | △ |
|---|---|---|---|---|---|---|---|---|---|
| | HierText | *SCUT-EnsText | *SCUT-Syn | | HierText | *TotalText | *FST | *TextSeg (*val*) | |
| *No Fine-tuning Baselines* | | | | | | | | | |
| MAE-VQGAN (Bar et al., 2022) | 28.52 / 32.71 | 29.12 / 44.58 | 27.25 / 45.81 | 28.30 / 41.70 | 1.93 | 5.83 | 6.57 | 13.54 | 6.97 |
| Painter (Wang et al., 2023b) | 22.68 / 47.17 | 26.29 / 52.08 | 24.07 / 54.60 | 24.35 / 51.28 | 4.08 | 6.01 | 4.88 | 9.70 | 6.17 |
| SegGPT (Wang et al., 2023d) | - | - | - | - | 3.12 | 9.58 | 9.45 | 25.36 | 11.88 |
| *Task-specific Fine-tuning (→) on HierText* | | | | | | | | | |
| Painter → *Rem.* | 26.14 / 31.09 | 36.15 / 21.37 | 33.85 / 29.30 | 32.05 / 27.92 | - | - | - | - | |
| SegGPT → *Seg.* | - | - | - | - | 60.60 | 65.10 | 59.12 | 75.75 | 65.14 |
| Painter → *Rem. + Seg.* | 28.17 / 24.76 | 36.48 / 21.05 | 34.38 / 28.38 | 32.34 / 24.73 | 64.72 | 67.81 | 61.09 | 77.02 | 67.16 |
| SegGPT → *Rem. + Seg.* | 28.16 / 25.51 | 36.56 / 21.19 | 34.42 / 28.32 | 33.05 / 24.34 | 65.23 | 68.53 | 62.20 | 77.40 | 68.34 |
| **ConText** | **39.48 / 6.35** | **37.67 / 12.87** | **37.93 / 13.91** | **38.36 / 11.04** | **74.86** | **78.02** | **71.02** | **82.31** | **76.77** |

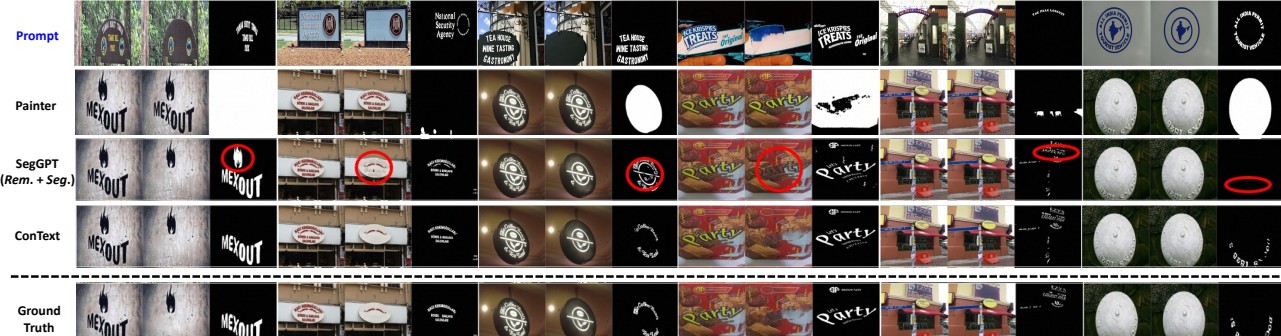

*Figure 4.* Visualized TotalText samples generated from several in-context generalists. Each set of samples consists of the (*original image, removal result, segmentation map*). The prompt refers to the given visual demonstration. The red circles indicate the differences in segmentation and removal results between our method and the fine-tuned SegGPT. Zoom in for a better view.

of-domain generalization capability. For task-specific fine-tuning methods, we strictly adhere to the corresponding fine-tuning settings. Clearly, our proposed method outperforms other models, achieving a PSNR of **38.36** and an FID of **11.04** across three text removal benchmarks, and an average fgIoU of **76.77%** for four text segmentation datasets. These results are significantly higher than those of non/task-specific fine-tuning baselines. The visualized results in Figure 4 further illustrate the superior in-context ability of our model, yielding better removal effects and segmentation masks compared to others.

**Comparison with Task-Specific Specialists.** Here, we present a stronger version **ConTextV** to conduct a comprehensive comparison against task-specific specialists. As shown in Table 2 & 3, **ConTextV** demonstrates an overwhelmingly superior performance in both text segmentation and removal tasks. In Table 2, with the support of just one randomly-selected demonstration, our method yields an average improvement of **+2.53%** in fgIoU, compared to other state-of-the-art (SOTA) methods. Notably, our approach achieves significant improvements on the previously unseen FST dataset, outperforming data-specific specialists. For text removal, as shown in Table 3, our method also achieves superior erasing effects compared to removal specialists in the SCUT-EnsText dataset. The related experimental results for SCUT-Syn can be found in Appendix B.3.

### 5.3. In-context Specificity

**In-context Learnability.** One unique attribute of in-context learning is its infer-by-prompt capability, yielding different levels of reasoning ability. In other words, an in-context model should be sensitive to the demonstration in terms of downstream task. Therefore, to verify this prompting flexibility, Figure 5 reports all models' performance given both the randomly-selected and ground-truth-based demonstration samples, yielding the upper and the normal in-context inference abilities. Particularly, as shown in this figure, all models are unable to implement flawless reconstruction even when being prompted by the ground-truth. Therefore, it is emphasized that the improvement of both the upper and lower performance is important to ICL models since current vision models are far from performing promising in-context learnability compared with those powerful LLMs. Based on these results, direct fine-tuning, as yielding significant improvement, could lead to invalid in-context learnability due to the minimal performance change regardless of the demonstration. However, our method exhibits a strong performance range, with an averaged difference of **+1.01** PSNR in text removal and **+5.39** fgIoU in segmentation. With the scaling of training benchmarks in our **ConTextV**, such a gap is further accentuated as the collective improvements of upper- and lower-performance. This indicates the model's capacity to adapt effectively even when demonstration sam-

*Table 2.* Comparison with the text segmentation **specialists** among four benchmarks. *FST dataset is not used for training in our model.

| Method | HierText | | TotalText | | *FST | | TextSeg | |
|---|---|---|---|---|---|---|---|---|
| | fgIoU↑ | F-score↑ | fgIoU↑ | F-score↑ | fgIoU↑ | F-score↑ | fgIoU↑ | F-score↑ |
| SegFormer (Xie et al., 2021a) | - | - | 73.31 | 0.846 | 60.44 | 0.753 | 84.59 | 0.916 |
| DeepLabV3+ (Chen et al., 2018) | - | - | 74.44 | 0.824 | 69.27 | 0.802 | 84.07 | 0.914 |
| HRNetV2-W48 (Wang et al., 2020) | - | - | 75.29 | 0.825 | 70.98 | 0.822 | 85.98 | 0.918 |
| HRNetV2-W48+OCR (Wang et al., 2020) | - | - | 76.23 | 0.832 | 72.45 | 0.830 | 85.98 | 0.918 |
| TexRNet + DeeplabV3+ (Xu et al., 2021) | - | - | 76.53 | 0.844 | 72.16 | 0.835 | 86.06 | 0.921 |
| TexRNet + HRNetV2-W48 (Xu et al., 2021) | 55.50 | 0.656 | 78.47 | 0.848 | 73.38 | 0.850 | 86.84 | 0.924 |
| TFT (Yu et al., 2023a) | - | - | 82.10 | 0.902 | 72.71 | 0.845 | 87.11 | 0.931 |
| EAFormer (Yu et al., 2024) | - | - | 82.73 | 0.906 | 72.63 | 0.840 | 88.06 | 0.939 |
| UPOCR (Peng et al., 2024b) | - | - | - | - | - | - | 88.76 | 0.940 |
| Hi-SAM (Ye et al., 2024) | 77.76 | 0.848 | 84.59 | 0.887 | - | - | 88.77 | 0.938 |
| **ConTextV** | **81.21** | **0.896** | **85.19** | **0.919** | **75.90** | **0.873** | **89.74** | **0.946** |

*Table 3.* Comparison with the removal **specialists**.

| Method | SCUT-EnsText | | | | | | |
|---|---|---|---|---|---|---|---|
| | PSNR↑ | MSSIM↑ | MSE↓ | AGE↓ | pEPs↓ | pCEPs↓ | FID↓ |
| Pix2Pix (Phillip et al., 2017) | 26.70 | 88.56 | 0.37 | 6.09 | 0.0480 | 0.0227 | 46.88 |
| STE (Nakamura et al., 2017) | 25.47 | 90.14 | 0.47 | 5.033 | 0.0533 | 0.0296 | 43.39 |
| EnsNeT (Zhang et al., 2019) | 29.54 | 92.74 | 0.24 | 4.16 | 0.0307 | 0.0136 | 32.71 |
| MTRNet++ (Tursun et al., 2020) | 29.63 | 93.71 | 0.23 | 3.51 | 0.0305 | 0.0168 | 35.50 |
| EraseNeT (Liu et al., 2020) | 32.30 | 95.42 | 0.15 | 3.02 | 0.0160 | 0.0090 | 19.27 |
| SSTE (Tang et al., 2021) | 35.34 | 96.24 | 0.09 | - | - | - | - |
| PSSTRNet (Lyu & Zhu, 2022) | 34.65 | 96.75 | 0.14 | 1.72 | 0.0135 | 0.0074 | - |
| CTRNet (Liu et al., 2022a) | 35.20 | 97.36 | 0.09 | 2.20 | 0.0106 | 0.0068 | 13.99 |
| GaRNet (Lee & Choi, 2022) | 35.45 | 97.14 | 0.08 | 1.90 | 0.0105 | 0.0062 | 15.50 |
| MBE (Hou et al., 2022) | 35.03 | 97.31 | - | 2.06 | 0.0128 | 0.0088 | - |
| PEN (Du et al., 2023c) | 35.72 | 96.68 | 0.05 | 1.95 | 0.0071 | 0.0020 | - |
| PERT (Wang et al., 2023e) | 33.62 | 97.00 | 0.13 | 2.19 | 0.0135 | 0.0088 | - |
| SAEN (Du et al., 2023a) | 34.75 | 96.53 | 0.07 | 1.98 | 0.0125 | 0.0073 | - |
| FETNet (Lyu et al., 2023) | 34.53 | 97.01 | 0.13 | 1.75 | 0.0137 | 0.0080 | - |
| ViTEraser (Peng et al., 2024a) | 36.87 | 97.51 | 0.05 | 1.72 | 0.0066 | 0.0035 | 10.15 |
| UPOCR (Peng et al., 2024b) | 37.14 | 97.62 | 0.04 | 1.72 | 0.0064 | 0.0034 | **10.47** |
| **ConTextV** | **40.83** | **98.76** | **0.03** | **0.76** | **0.0053** | **0.0029** | 11.63 |

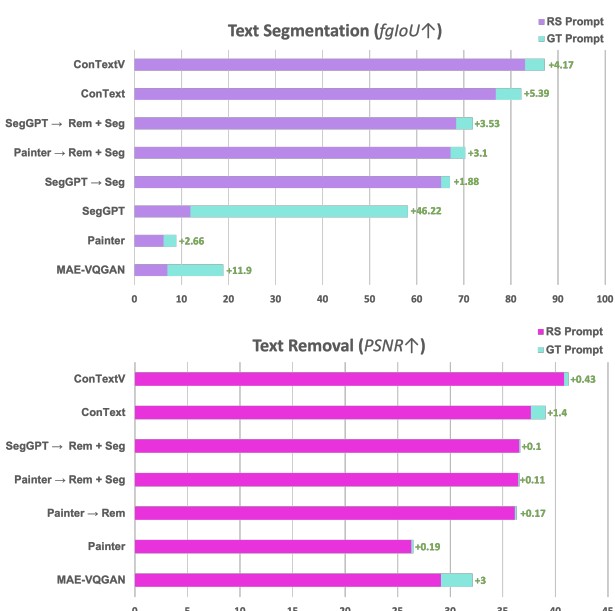

*Figure 5.* Comparison of several visual generalists on text segmentation and removal tasks when given the *randomly-selected* (RS) and *ground-truth-based* (GT) prompts. Here the segmentation (removal) task is evaluated against four (SCUT-EnsText) benchmarks. This two-case performance range denotes a model's substantial in-context learnability towards these tasks, where a sounding upper and lower bounds indicates its strong scalability potential.

ples differ from the ground-truth. Overall, the proposed model exhibits both strong upper and lower performance bounds compared to other methods, highlighting its scalability and versatility. Appendix B.2 presents the specific numerical results for each benchmark, which intuitively demonstrate our model's powerful in-context learnability.

**In-Context Understanding.** As claimed above, one of the amazing advantages of ICL is providing a flexible user-oriented interaction with models. To further evaluate the generalized inference capability of our model towards the given demonstration, we specifically construct a dataset with explicit visual markers, namely **PromptText**, including randomly-colored circle, stroke, and box, to mimic the user behavior on demonstration to segment and erase as required (please refer to Appendix B.4 for more details about the construction of this dataset). Particularly, such a dataset, merely serving as an evaluation benchmark, is *not training-involved*. Table 4 shows the results of this prompting dataset among several methods, and clearly, our method has achieved an overall promising performance when compared to all other methods. For those specialists, with reflexively performing text segmentation/removal, their low performance is reasonable due to the lack of understanding towards the explicit prompting. The visual generalists, despite having limited

prompt comprehension, also exhibit subpar text recognition capabilities. In contrast, our methods surprisingly demonstrate a thorough understanding of these explicit prompts, resulting in superior segmentation and removal performance (as shown in Figure 6). This experiment also highlights the value of exploring visual in-context inference for text recognition, driving a more adaptable form of user-model interaction. Besides, we argue that this experiment also reveals the visual-cues prompting ability of our model. As stated in Shtedritski et al. (2023); Yang et al. (2023), an emerging ability of recognizing explicit visual hints has been explored for current foundation models, enhancing the fine-grained and localized recognition capability through

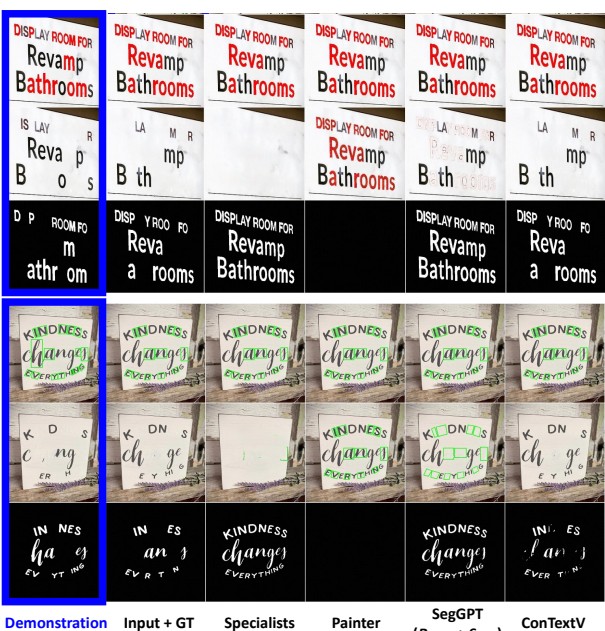

*Figure 6.* Visualized samples of several methods on the prompting datasets. The results of the specialists are obtained through Hi-SAM and ViT-eraser, respectively. Other generalists are prompted by the given demonstration. Zoom in for a better view.

*Table 4.* Comparison with several frameworks on our designed user-prompted datasets. Note that all these explicitly-prompting datasets are **not used during the training** for all methods, which evaluates a model's in-context generalized understanding.

| Method | PromptText *Rem.* | | | PromptText *Seg.* | |
|---|---|---|---|---|---|
| | PSNR↑ | MSSIM↑ | FID↓ | fgIoU↑ | F-score↑ |
| *Task-specific Specialists* | | | | | |
| Hi-SAM (Ye et al., 2024) | - | - | - | 43.51 | 0.621 |
| ViT-Eraser (Peng et al., 2024a) | 20.23 | 90.25 | 112.81 | - | - |
| *Visual In-context Generalists* | | | | | |
| Painter (Wang et al., 2023b) | 19.63 | 88.91 | 81.11 | 7.50 | 0.140 |
| Painter (Wang et al., 2023b) → *Rem.* | 24.11 | 90.32 | 68.02 | - | - |
| SegGPT (Wang et al., 2023d) → *Seg.* | - | - | - | 43.88 | 0.610 |
| SegGPT (Wang et al., 2023d) → *Seg. + Rem.* | 23.73 | 90.31 | 68.70 | 44.69 | 0.618 |
| **ConText** | 33.16 | 98.08 | 41.24 | 54.74 | 0.708 |
| **ConTextV** | **38.14** | **99.06** | **33.59** | **59.19** | **0.744** |

a simple but explicit visual marker on the query object. In conclusion, our model has demonstrated an exceptional training-free generalized and recognition ability.

### 5.4. Ablation Studies

In this section, we will make relevant ablations about our method, such as the effectiveness of our designed module, performance with multi-demonstration, and double in-context inference. Unless otherwise specified, the overall ablations are conducted by using **ConText** (trained with HierText).

**Effectiveness of Individual Module.** Table 5 presents an ablation study assessing the effectiveness of various design components of our model. Here the *baseline* refers to the multi-task fine-tuning version of SegGPT (SegGPT → *Rem.* + *Seg.*). As shown in Table 5, the intuitive *linear fusion*

*Table 5.* Effectiveness of designed items on our method. The segmentation (removal) is evaluated via TotalText (SCUT-ENS) based on fgIoU (PSNR). RS (GT) denotes the model's performance with randomly-selected (ground-truth) demonstration.

| Baseline | Linear Fusion ($\widetilde{\mathbf{F}}_1$) | CAA ($\widetilde{\mathbf{F}}_1 + \widetilde{\mathbf{F}}_2$) | SP-0.2 | SP-0.6 | *Seg.* RS / GT | *Rem.* RS / GT |
|---|---|---|---|---|---|---|
| ✔ | | | | | 68.53 / +1.57 | 34.42 / +0.17 |
| ✔ | ✔ | | | | 72.14 / +1.08 | 35.75 / +0.41 |
| ✔ | | ✔ | | | **79.14** / +0.65 | **38.59** / +0.37 |
| ✔ | | ✔ | ✔ | | 78.02 / +3.98 | 37.67 / +1.42 |
| ✔ | | ✔ | | ✔ | 77.14 / **+5.83** | 36.12 / **+2.13** |

*Table 6.* Effectiveness of the masking ratio value (%) on our proposed method. RS (GT) denotes the model's performance with randomly-selected (ground-truth) demonstration.

| Masking Ratio (%) | TotalText *Seg.* RS / GT | SCUT-Ens *Rem.* RS / GT |
|---|---|---|
| 25 | 75.80 / +2.34 | 36.15 / +1.39 |
| 35 | 76.23 / +3.02 | 36.67 / +1.72 |
| 55 | 77.45 / +2.56 | 36.89 / +1.53 |
| 75 | 77.74 / +2.84 | 37.21 / +1.62 |
| 85 | **78.02** / **+3.98** | **37.67** / **+1.68** |
| 95 | 78.04 / +3.04 | 36.83 / +1.29 |

($\widetilde{\mathbf{F}}_1$) yields a significant improvement (**+3.61%** fgIoU for segmentation and **+1.33** PSNR for removal) on both downstream tasks, demonstrating the benefits of context fusion. Based on this, our enhanced context fusion could drastically improve the model's performance by achieving an elation of **+10.61%** fgIoU and **+4.17** PSNR, which strongly verifies the superiority of our proposed method. However, similar improvements have not been observed in our model when given the ground-truth as the demonstration. Different from the natural object-level recognition, the text recognition, though comprised of merely binary units, is difficult to define its visually homogeneous counterpart. Therefore, without the guidance from proper demonstration, these designed modules shall drive the original in-context model into a pure task-specific specialists. As shown in this table, merely a marginal fluctuation is observed between the random and ground-truth demonstration. By introducing the *self-prompting* (SP) manner, the model tends to maintain both the task-specific capacity and in-context generalization, while the over-usage of such a mechanism would degrade the model's task-specific performance due to the reduced demonstration diversity. This finding also highlights the importance of balancing demonstration diversity to optimize model outcomes in in-context learning scenarios.

**Effectiveness of Masking Ratio.** We have conducted ablations regarding the masking ratio at wide range ($25\%-95\%$). As shown in Table 6, training with lower mask ratio would lead to a certain decrease on both the removal and segmentation tasks, especially under the $25\%$ case. With the growing number of erased parts, the model tends to show consistent improvements on both downstream tasks, reaching the

*Table 7.* Performance of our proposed **ConText** under different randomly-selected demonstration number against 3 benchmarks.

| Demonstration Number | TotalText *Seg.* | TextSeg (*val*) *Seg.* | SCUT-Ens *Rem.* |
|---|---|---|---|
| | fgIoU (↑) | fgIoU (↑) | PSNR (↑) |
| *Multi-demonstration Inference* | | | |
| 1 (*Baseline*) | 78.02 | 82.31 | 37.67 |
| 3 | 78.12 (+0.10) | 82.47 (+0.16) | 37.98 (+0.31) |
| 5 | 78.64 (+0.62) | 82.83 (+0.54) | 38.45 (+0.78) |
| *Double Inference* | | | |
| 1 | 78.26 (+0.25) | 82.86 (+0.55) | 38.11 (+0.44) |

*Table 8.* Computational efficiency of the designed items on our method. The results are evaluated on HierText, and FLOPs and inference time are calculated by forwarding one $512 \times 512$ image on one A100, with the inference time reported in seconds (sec) and training time reported in minutes (min) per epoch.

| Method | Training Time | Inference Time | Model FLOPs |
|---|---|---|---|
| Baseline | 3.8 min | 0.09 sec | 666.76G |
| Baseline + SP | 4.2 min | 0.09 sec | 666.76G (0%) |
| Baseline + CAA | 4.6 min | 0.12 sec | 683.96G (+2%) |
| Baseline + CAA + SP | 4.8 min | 0.12 sec | 683.96G (+2%) |

peak point with $85\%$ masking ratio. However, beyond that masking proportion, the model showcases an evident performance decline. These results align with the conclusions of Fang et al. (2023), confirming the effectiveness of the proper application of the masking strategy.

**Multi-demonstration/Double Inference.** Table 7 presents the effectiveness of multi-demonstration- and double-inference on our proposed framework. To achieve the former one, we follow the feature ensemble operation in Wang et al. (2023d) that first feeds the different demonstration-query pairs as a batch-wise forward process, and then averagely fuse the query features at the specific each layer of the model (specific implementation could refer to Wang et al. (2023d)). As shown in this table, fusing the multi-demonstration could yield an overall improved performance when compared to normal 1-shot inference. However, it is observed that this use of multi-shot does not yield as much improvement as reported by Wang et al. (2023d), likely due to the heterogeneous visual attributes involved in text recognition. Besides, the introduce of multi-demonstration would increase the labeling efforts for both segmentation and removal tasks. Therefore, there exists a trade-off between model accuracy and the cost of data labeling. Another interesting trial of our inference manner is to use the first-time generated results as the new demonstration to perform a second-time in in-context inference, and such a *double inference* is also similar to a kind of self-training. Table 7 showcases the effectiveness of this inference mechanism, which also brings a certain degree of improvement to the model's performance. However, considering the computational costs, we do not adopt this approach for relatively minor improvements.

**Computational Efficiency.** Table 8 reports the additional costs of our designed modules. Clearly, we find that SP

*Table 9.* Performance comparison on CLWD (Liu et al., 2021).

| Method | *Rem.* (PSNR) | *Seg.* (fgIoU) |
|---|---|---|
| SegGPT | 30.11 | 74.42 |
| PFMNet (Niu et al., 2023) | 39.45 | 79.09 |
| **ConText** | **40.73** | **82.16** |

incurs an additional training burden of $+0.4$ minutes per epoch. However, this cost is deemed acceptable due to the moderate engagement of SP (SP-0.2) during training. Moreover, SP is not utilized during inference, yielding no additional computational burden for inference. Furthermore, CAA introduces extra computational costs during both training and inference. However, as a lightweight cross-attention module, it only increases model complexity by a manageable $2\%$. Consequently, it leads to a mere increase of $+0.03$ seconds ($+0.8$ minutes/epoch) during inference (training). Based on this, we can safely conclude these modules indicate a reasonable level of computational efficiency.

**Beyond OCR.** To verify the generalization of our task-in-chain concept, we (following similar training strategy) have additionally explored our ConText on one prevailing watermark removal benchmark, CLWD (Liu et al., 2021). Table 9 reports the results of SegGPT and a leading specialist (Niu et al., 2023). Clearly, our approach demonstrates superior performance, with achieving advanced performance among both removal and segmentation tasks, which further validating the task-wise generalizability of ConText.

## 6. Conclusion

In this paper, we presented, to the best of our knowledge, the first exploration of establishing a *visual in-context learning* (V-ICL) paradigm for fine-grained text recognition tasks, including text segmentation and removal. To achieve this, we sought a *single-task-targeted* baseline solution based on the prevailing V-ICL frameworks, which typically regulates in-context inference as a query-label-reconstruction process. Beyond simple task-specific fine-tuning, we proposed an end-to-end in-context generalist elicited from a *task-chaining* prompt that explicitly chaining up tasks as one enriched demonstration, leveraging inter-task correlations to improve the in-context reasoning capabilities. Additionally, we introduced the *context-aware aggregation* (CAA) module and *self-prompting* (SP) training techniques to further strengthen the model's understanding of in-context representations, resulting in a significant enhancement of reasoning on heterogeneous visual patterns. Through quantitative and qualitative experiments, we demonstrated the grounding effectiveness and superiority of our framework across various in-domain and out-of-domain text recognition tasks, outperforming both current generalists and specialists. Overall, we hope this pioneering work will encourage further development of V-ICL in text recognition.

## Impact Statement

Note that all our training datasets, and the data for training our framework are sourced from the Internet. Consequently, the collection of these datasets raises concerns regarding privacy if not appropriately regulated. Additionally, the stroke and removal labels heavily rely on human annotators or the off-the-shelf tools, which can introduce potential noises and biases, intentional or unintentional, if the annotators are not impartial. It is key to address these issues through proper data regulation, privacy protection measures, and meticulous selection of the annotated information to ensure fairness and relieve potential biases.

## Acknowledgements

This work is supported by the National Key R&D Program of China (No. 2022ZD0160703), National Natural Science Foundation of China (No. 62306178), and STCSM (No. 22DZ2229005), 111 plan (No. BP0719010). This work is also supported by Alibaba Research Intern Program.

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

# A. Motivations

## A.1. Pilot Experiments

Similar to natural image segmentation (Zhang et al., 2021; Li et al., 2023a;b; Zhang et al., 2023a; Ma et al., 2023; Yang et al., 2024; Zhou et al., 2024; Liu et al., 2024; Zhang et al., 2025), we believe that text segmentation follows similar inherent pattern learning. Recall that in Section 4 we conduct a simple pilot experiment to validate the feasibility of our proposed task-chaining demonstration. Table 10 reports the results of using this simple demonstration recasting. Specifically, we fine-tune the baseline pipeline Painter (Wang et al., 2023b) and SegGPT (Wang et al., 2023d) by forwarding this new designed demonstration, and then only reconstruct merely one type of task during the training-inference (All features are still directly fused to one label representation). Similarly, merely the training target task would be evaluated during the inference, where the other task prompt is served with ground-truth label. For instance, **Rem-based Seg.** refers to reconstruct the segmentation task by introducing the removal label unchanged as the ground-truth for both query and demonstration pair, which is also provided during the inference. As shown in this table, the observed improvement experimentally validate the potential superiority of utilizing task-level connection to improve model's ICL ability.

*Table 10.* Pilot experiment regarding the motivation of task-chaining. The training dataset is HierText.

| Method | TotalText *Seg.* | SCUT-Ens *Rem.* |
|---|---|---|
| | fgIoU (↑) | PSNR (↑) |
| *Task-specific Fine-tuning* | | |
| Painter → *Rem.* | - | 36.15 |
| SegGPT → *Seg.* | 60.60 | - |
| *Task-Chaining* | | |
| *Seg-based Rem.* (Painter) | - | 37.02 (+0.87) |
| *Rem-based Seg.* (SegGPT) | 63.22 (+2.62) | - |

## A.2. Label role in ICL

We design a simple cross-attention-based architecture to reinforce the label representation in Section 4. This design is motivated by Wang et al. (2023f); Yu & Ananiadou (2024), where they validated that label position could serve as an anchor to absorb the pattern of the provided demonstration. Figure 7 shows an illustrative explanation for this hypothesis, and it is clearly seen that the label position plays a vital role in understanding the demonstration. Besides, the final input text (last position) should have the most comprehensive pattern for all the demonstrations. Therefore, we could find that the linear fusion in Wang et al. (2023b;d) is quite reasonable for the purpose of merging the demonstration information for the final label. we conduct a simple pilot experiment to validate the feasibility of our proposed task-chaining demonstration.

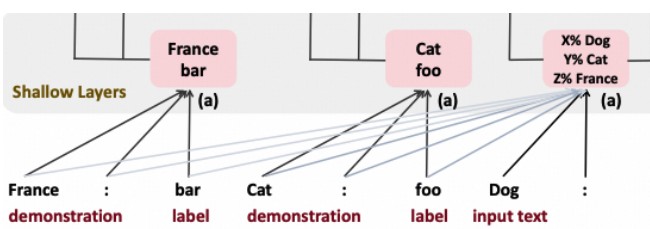

*Figure 7.* Illustration of label role proposed in Yu & Ananiadou (2024). Intuitively, shallow layers merge features into label positions and last position within ICL. This mechanism also inspires the design of our context-aware module.

# B. Experiments

## B.1. More Training Details

Our model adopts *vision transformers* (ViT) (Dosovitskiy et al., 2020) as the backbone. We use the pre-trained checkpoint from Wang et al. (2023d) as the initialization. Specifically, our model is built on ViT-L + decoder architecture, which is the same as Wang et al. (2023b;d). Notably, our proposed method requires two kinds of labels for the training datasets, i.e., stroke

masks and text-removed images. For HierText and TotalText, we directly adopt the off-the-shelf human-evaluated removal labels from (Zhu et al., 2024). For TextSeg, we turn to a promising text-eraser method (Peng et al., 2024a) to generate the corresponding removal labels. Note that most removal-targeted works (Tursun et al., 2020; Peng et al., 2024a) would train and evaluate SCUT-Syn, which is a synthetic benchmark. To align with them, we also make a similar comparison, and relevant discussion is presented at Appendix B.3. Compared to **ConText**, we use the SCUT-EnsText as one of the additional training benchmarks during the fine-tuning stage. To make their corresponding segmentation labels, we refine and expand upon the labeling techniques described in Peng et al. (2024a) by computing the distance between two images in gray-scale mode, thereby generating better fine-grained mask. Such a method is also applied on the following SCUT-Syn benchmarks. However, such self-generated labels, accompanied by label noises, are accurately inferior to those human-annotated masks in segmentation benchmarks. Therefore, to guarantee the training stability of **ConTextV**, we firstly train the model by using those three segmentation benchmarks (HierText + TextSeg + TotalText), and then fine-tune the model with all of the training datasets (HierText + TextSeg + TotalText + SCUT-EnsText) with merely 2 epochs. The whole training time takes 8 (12) hours for Context (**ConTextV**). To guarantee a promising fine-grained word-level recognition in the generative paradigm, we follow (Chen et al., 2023) additionally introduce the cross-entropy-based pixel-level supervision $\mathcal{L}_{\text{pix}}$, accompanied with an extra merely-training-available decoder. The weight for the removal reconstruction loss is set to 0.3, and 1 for the pixel-level supervision loss $\mathcal{L}_{pix}$. The removal reconstruction loss is adopted as smooth-l1 for both reconstructing the segmentation mask and removal image. The probability of self-prompting is set to 0.2. The simple light-weight decoder for pixel-level supervision, comprised of two convolution layers, is **not used** during the inference stage. All datasets used in our paper are described as follows:

1. **HierText**: A fine-grained real-world segmentation benchmark, including 8,281 training samples, 1,724 validation samples, and 1,634 test samples. We use all the training samples during the training stage and evaluate the model with the validation set.

2. **TextSeg**: A large-scale fine-annotated text segmentation dataset with 4,024 images of scene text and design text. The training, validating, and testing sets contain 2,646, 340, and 1,038 samples, respectively.

3. **TotalText**: A prevailing small-scale text segmentation dataset. The training and validating sets contain 1,255, and 300, respectively.

4. **FST**: A prevailing small-scale English text segmentation dataset. The training and validating sets contain 229, and 233, respectively. Besides, the annotation of FST is coarse equipped with part patch-like foreground regions.

5. **SCUT-EnsText**: is a real-world scene text removal dataset, comprising 2,749 samples for training and 813 samples for testing.

6. **SCUT-Syn**: is a purely synthetic scene text removal dataset, comprising 8,000 samples for training and 800 samples for testing.

## B.2. Detailed Results of In-context Learnability

Here we present the detailed numeric results of our performance against each benchmark under both the randomly-selected and the ground-truth demonstration. As shown in Table 11,12 and 13, compared to other methods, our model could demonstrate promising in-context learnability with a notable upper-and-lower performance gap. Besides, the numerical comparison with other specialists shows a huge room for further improvement of our method.

## B.3. SCUT-Syn Evaluation

Here we present our performance against SCUT-Syn benchmark. Compared to those data-specific methods, as shown in Table 13, our method could achieve comparable performance on this synthetic dataset. Note that in Table 3, our proposed method is unable to reach the best performance against those specialists, and we speculate such a suboptimal performance may attribute to the synthetic-natural training domain gap. To verify this, Table 14 highlights the impact of incorporating the SCUT-Syn synthetic dataset on the model's performance in segmentation and removal tasks. Notably, using only the synthetic data allows the model to achieve strong in-domain performance, equally the state-of-the-art results with 0.01 MSE. However, this comes at the cost of reduced generalization to other datasets. Conversely, without any synthetic data, the model performs well on natural datasets. As more synthetic data is integrated, the model's performance shifts, balancing

*Table 11.* Comparison with different V-ICL frameworks against several text removal (*Rem.*) and text segmentation (*Seg.*) benchmarks. The upper in-context inference performance is marked (the demonstration is the ground-truth). The performance range denotes a model's substantial in-context learnability towards these tasks, where a sounding upper and lower bounds indicates its strong scalability potential.

| Method | Text Removal (PSNR↑ / FID↓) | | | △ | Text Segmentation (fgIoU ↑) | | | | △ |
|---|---|---|---|---|---|---|---|---|---|
| | HierText | *SCUT-EnsText | *SCUT-Syn | | HierText | *TotalText | *FST | *TextSeg (val) | |
| *No Fine-tuning Baselines* | | | | | | | | | |
| MAE-VQGAN (Bar et al., 2022) | 28.52 / 32.71 | 29.12 / 44.58 | 27.25 / 45.81 | 28.30 / 41.70 | 1.93 | 5.83 | 6.57 | 13.54 | 6.97 |
| | 30.28 / 27.69 | 32.12 / 38.01 | 30.49 / 36.31 | +2.66 / -7.70 | 8.82 | 17.86 | 20.51 | 28.30 | +11.90 |
| Painter (Wang et al., 2023b) | 22.68 / 47.17 | 26.29 / 52.08 | 24.07 / 54.60 | 24.35 / 51.28 | 4.08 | 6.01 | 4.88 | 9.70 | 6.17 |
| | 23.20 / 46.08 | 26.48 / 51.91 | 24.03 / 54.62 | +0.22 / -0.41 | 4.18 | 14.47 | 4.94 | 9.74 | +2.66 |
| SegGPT (Wang et al., 2023d) | - | - | - | - | 3.12 | 9.58 | 9.45 | 25.36 | 11.88 |
| | - | - | - | - | 41.28 | 62.06 | 60.92 | 70.12 | +46.22 |
| *Task-specific Fine-tuning (→) on HierText* | | | | | | | | | |
| Painter → *Rem.* | 26.14 / 31.09 | 36.15 / 21.37 | 33.85 / 29.30 | 32.05 / 27.92 | - | - | - | - | |
| | 26.29 / 30.90 | 36.32 / 20.47 | 35.62 / 27.09 | +0.69 / -1.76 | - | - | - | - | |
| SegGPT → *Seg.* | - | - | - | - | 60.60 | 65.10 | 59.12 | 75.75 | 65.14 |
| | - | - | - | - | 62.91 | 66.13 | 63.56 | 77.48 | +1.88 |
| Painter → *Rem. + Seg.* | 28.17 / 24.76 | 36.48 / 21.05 | 34.38 / 28.38 | 32.34 / 24.73 | 64.72 | 67.81 | 61.09 | 77.02 | 67.16 |
| | 29.18 / 20.71 | 36.59 / 20.88 | 34.53 / 28.13 | +1.09 / -1.49 | 66.97 | 69.99 | 65.09 | 78.99 | +3.10 |
| SegGPT → *Rem. + Seg.* | 28.16 / 25.51 | 36.56 / 21.19 | 34.42 / 28.32 | 33.05 / 24.34 | 65.23 | 68.53 | 62.20 | 77.40 | 68.34 |
| | 28.18 / 21.49 | 36.66 / 20.94 | 34.59 / 28.14 | +0.09 / -0.82 | 67.97 | 70.15 | 66.71 | 80.65 | +3.53 |
| **ConText** | **39.48 / 6.35** | **37.67 / 12.87** | **37.93 / 13.91** | **38.36 / 11.04** | **74.86** | **78.02** | **71.02** | **82.31** | **76.77** |
| | **39.68 / 6.08** | **39.07 / 12.30** | **39.35 / 13.46** | **+1.01 / -0.43** | **78.12** | **82.01** | **80.29** | **87.35** | **+5.39** |

*Table 12.* Comparison with the text segmentation **specialists** among four benchmarks. *FST dataset is not used for training in our model.

| Method | HierText | | TotalText | | *FST | | TextSeg | |
|---|---|---|---|---|---|---|---|---|
| | fgIoU↑ | F-score↑ | fgIoU↑ | F-score↑ | fgIoU↑ | F-score↑ | fgIoU↑ | F-score↑ |
| SegFormer (Xie et al., 2021a) | - | - | 73.31 | 0.846 | 60.44 | 0.753 | 84.59 | 0.916 |
| DeepLabV3+ (Chen et al., 2018) | - | - | 74.44 | 0.824 | 69.27 | 0.802 | 84.07 | 0.914 |
| HRNetV2-W48 (Wang et al., 2020) | - | - | 75.29 | 0.825 | 70.98 | 0.822 | 85.98 | 0.918 |
| HRNetV2-W48+OCR (Wang et al., 2020) | - | - | 76.23 | 0.832 | 72.45 | 0.830 | 85.98 | 0.918 |
| TexRNet + DeeplabV3+ (Xu et al., 2021) | - | - | 76.53 | 0.844 | 72.16 | 0.835 | 86.06 | 0.921 |
| TexRNet + HRNetV2-W48 (Xu et al., 2021) | 55.50 | 0.656 | 78.47 | 0.848 | 73.38 | 0.850 | 86.84 | 0.924 |
| TFT (Yu et al., 2023a) | - | - | 82.10 | 0.902 | 72.71 | 0.845 | 87.11 | 0.931 |
| EAFormer (Yu et al., 2024) | - | - | 82.73 | 0.906 | 72.63 | 0.840 | 88.06 | 0.939 |
| UPOCR (Peng et al., 2024b) | - | - | - | - | - | - | 88.76 | 0.940 |
| Hi-SAM (Ye et al., 2024) | 77.76 | 0.848 | 84.59 | 0.887 | - | - | 88.77 | 0.938 |
| **ConTextV** | **81.21** | **0.896** | **85.19** | **0.919** | **75.90** | **0.873** | **89.74** | **0.946** |
| | 83.67 | 0.911 | 88.13 | 0.937 | 82.98 | 0.907 | 93.95 | 0.969 |

between in-domain excellence and generalization. The optimal configuration was found by using 25% of the training samples from SCUT-Syn, achieving a comprehensive performance balance. This underscores the domain gap issue between synthetic and natural data, emphasizing the importance of an appropriate data mix for optimal results.

## B.4. PromptText

Recall that in Section 5.3 we introduce a self-designed dataset to mimic the human-based instructed prompts on the textual images. To this end, we select the validation set from TextSeg, and adopts its original annotation to make the corresponding explicit prompts, which roughly contains:

1. Select the erasing probability from $\{0.3, 0.5, 0.7\}$, and such a probability is used for deciding whether the annotation is erased.

2. Select the annotation type, which contains stroke-level, box-level, and circle-level. The circle-level annotation could be generated from depiction of a circumscribed circle highlighted by a box annotation.

3. Select the color for this prompt from Red, Green, and Blue.

*Table 13.* Comparison with the **specialists** tailored for text removal among two benchmarks. Note that compared to other methods, our framework is not trained with *SCUT-Syn datasets. Particularly, SCUT-Syn is an artificially synthesized dataset.

| Method | SCUT-EnsText | | | | | | | *SCUT-Syn | | | | | |
|---|---|---|---|---|---|---|---|---|---|---|---|---|---|
| | PSNR↑ | MSSIM↑ | MSE↓ | AGE↓ | pEPs↓ | pCEPs↓ | FID↓ | PSNR↑ | MSSIM↑ | MSE↓ | AGE↓ | pEPs↓ | pCEPs↓ |
| Pix2Pix (Phillip et al., 2017) | 26.70 | 88.56 | 0.37 | 6.09 | 0.0480 | 0.0227 | 46.88 | 26.76 | 91.08 | 0.27 | 5.47 | 0.0473 | 0.0244 |
| STE (Nakamura et al., 2017) | 25.47 | 90.14 | 0.47 | 5.033 | 0.0533 | 0.0296 | 43.39 | 25.40 | 90.12 | 0.65 | 9.49 | 0.0553 | 0.0347 |
| EnsNeT (Zhang et al., 2019) | 29.54 | 92.74 | 0.24 | 4.16 | 0.0307 | 0.0136 | 32.71 | 37.36 | 96.44 | 0.21 | 1.73 | 0.0069 | 0.0020 |
| MTRNet++ (Tursun et al., 2020) | 29.63 | 93.71 | 0.23 | 3.51 | 0.0305 | 0.0168 | 35.50 | 34.55 | 98.45 | 0.04 | - | - | - |
| EraseNeT (Liu et al., 2020) | 32.30 | 95.42 | 0.15 | 3.02 | 0.0160 | 0.0090 | 19.27 | 38.32 | 97.67 | 0.02 | 1.60 | 0.0048 | 0.0004 |
| SSTE (Tang et al., 2021) | 35.34 | 96.24 | 0.09 | - | - | - | - | 38.60 | 97.55 | 0.02 | - | - | - |
| PSSTRNet (Lyu & Zhu, 2022) | 34.65 | 96.75 | 0.14 | 1.72 | 0.0135 | 0.0074 | - | 39.25 | 98.15 | 0.02 | 1.20 | 0.0043 | 0.0008 |
| CTRNet (Liu et al., 2022a) | 35.20 | 97.36 | 0.09 | 2.20 | 0.0106 | 0.0068 | 13.99 | 41.28 | 98.52 | 0.02 | 1.33 | 0.0030 | 0.0007 |
| GaRNet (Lee & Choi, 2022) | 35.45 | 97.14 | 0.08 | 1.90 | 0.0105 | 0.0062 | 15.50 | - | - | - | - | - | - |
| MBE (Hou et al., 2022) | 35.03 | 97.31 | - | 2.06 | 0.0128 | 0.0088 | - | 43.85 | 98.64 | - | 0.94 | 0.0013 | 0.00004 |
| PEN (Du et al., 2023c) | 35.72 | 96.68 | 0.05 | 1.95 | 0.0071 | 0.0020 | - | 38.87 | 97.83 | 0.03 | 1.38 | 0.0041 | 0.0004 |
| PERT (Wang et al., 2023e) | 33.62 | 97.00 | 0.13 | 2.19 | 0.0135 | 0.0088 | - | 39.40 | 97.87 | 0.02 | 1.41 | 0.0046 | 0.0007 |
| SAEN (Du et al., 2023a) | 34.75 | 96.53 | 0.07 | 1.98 | 0.0125 | 0.0073 | - | 38.63 | 98.27 | 0.03 | 1.39 | 0.0043 | 0.0004 |
| FETNet (Lyu et al., 2023) | 34.53 | 97.01 | 0.13 | 1.75 | 0.0137 | 0.0080 | - | 39.14 | 97.97 | 0.02 | 1.26 | 0.0046 | 0.0008 |
| ViTEraser (Peng et al., 2024a) | 36.87 | 97.51 | 0.05 | 1.72 | 0.0066 | 0.0035 | 10.15 | 42.97 | 98.55 | 0.01 | 1.11 | 0.0015 | 0.000011 |
| UPOCR (Peng et al., 2024b) | 37.14 | 97.62 | 0.04 | 1.72 | 0.0064 | 0.0034 | **10.47** | - | - | - | - | - | - |
| **ConTextV** | **40.83** | **98.76** | **0.03** | **0.76** | **0.0053** | **0.0029** | 11.63 | 38.30 | 98.30 | 0.07 | 0.99 | 0.0049 | 0.0032 |
| | 41.26 | 98.86 | 0.02 | 0.72 | 0.0047 | 0.0025 | 10.82 | 38.65 | 98.37 | 0.06 | 0.94 | 0.0044 | 0.0029 |

*Table 14.* Performance with varying proportions of training samples from SCUT-Syn. The '+' symbol indicates the number of SCUT-Syn training samples added to the baseline training dataset used in **ConTextV**. RS (GT) denotes the model's performance with randomly-selected (ground-truth) demonstration.

| SCUT-Syn Training Data Volume | SCUT-Syn *Rem.* (RS) | TotalText *Seg.* | SCUT-Ens *Rem.* |
|---|---|---|---|
| | MSE ↓ / PSNR ↑ | RS / GT | RS / GT |
| ONLY SCUT-Syn (100%) | 0.01 / 43.14 | 62.15 / +3.33 | 36.78 / +0.76 |
| NO SCUT-Syn (0%) | 0.07 / 38.30 | 85.19 / +2.94 | 40.83 / +0.43 |
| + 2,000 (25%) | 0.04 / 39.53 | 85.02 / +2.68 | 40.33 / +0.56 |
| + 4,000 (50%) | 0.04 / 39.82 | 84.73 / +2.37 | 40.06 / +0.43 |
| + 8,000 (100%) | 0.03 / 40.07 | 83.19 / +2.45 | 39.47 / +0.62 |

4. Mark each image with the selected color and annotation type for each no-erased label, and generate the corresponding removal and segmentation image.

This dataset contains 429 samples, and each image has 3-level annotation based on the erasing probability. Some visualized samples are shown in Figure 8. Note that this dataset is merely used for evaluation.

### B.5. More Visualized Results

Figure 9 presents a visual comparison between our method and other SOTA specialists. The enhanced and fine-grained segmentation and erasing detail highlights the superiority and effectiveness of our proposed **ConTextV** in addressing these tasks. Particularly, our model could even achieve better results than the given ground-truth label.

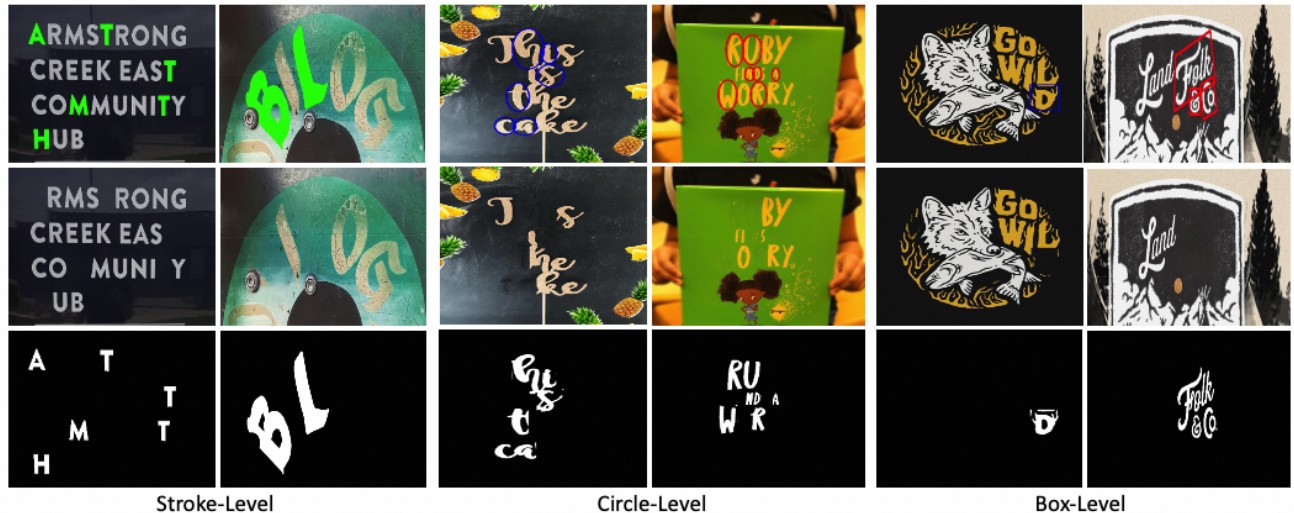

Figure 8. Visualized samples of our designed PromptText. Zoom in for a better view.

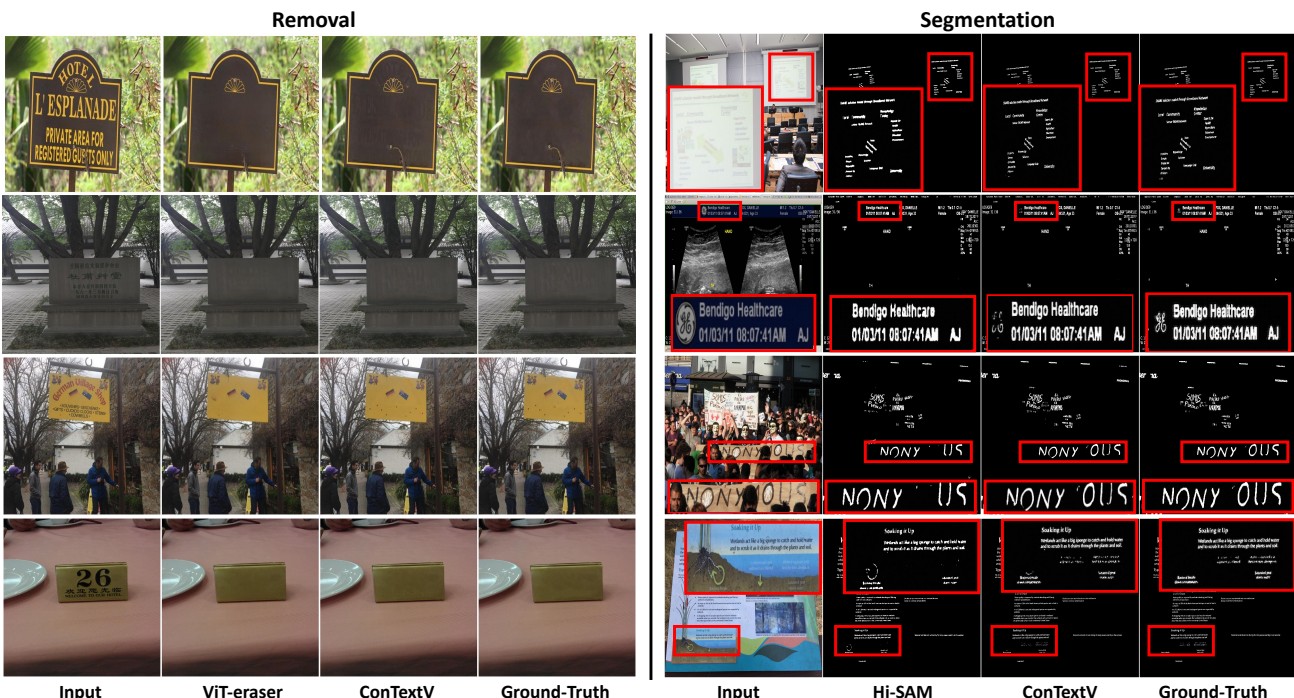

Figure 9. Qualitative comparison of our method and existing specialists on SCUT-EnsText and HierText. Our method is prompted by random demonstration. Clearly, our framework demonstrates promising performance across these tasks. Zoom in for a better view.

