# OpenReview forum: "ConText: Driving In-context Learning for Text Removal and Segmentation"
_ICML.cc/2025/Conference — ICML 2025 poster_

### Official Review · Reviewer_qCFq · 2025-03-11

**Overall Recommendation:** 2

**Summary:**

This paper introduces ConText, a visual in-context learning framework tailored for OCR tasks such as text removal and text segmentation.
It employs task-chaining, context-aware aggregation, and a self-prompting strategy to leverage multi-task logic and enhance in-context reasoning. Extensive experiments demonstrate state-of-the-art performance on multiple benchmarks, confirming its effectiveness and adaptability.

**Claims And Evidence:**

The paper's claims are generally supported by extensive experimental evidence across several benchmarks.

**Essential References Not Discussed:**

The paper appears to have sufficiently discussed all key related works, with no significant essential references omitted.

**Experimental Designs Or Analyses:**

The experimental designs and analyses, including ablation studies and cross-benchmark comparisons, are generally sound.

**Methods And Evaluation Criteria:**

The proposed methods and evaluation criteria are logically appropriate for addressing OCR tasks within a visual in-context learning framework.

**Other Comments Or Suggestions:**

1.The related work section is weakly written; for example, the description "Unlike these discriminative models, this paper proposes a universally generative-based framework for the task." does not clearly outline the shortcomings of prior works before introducing the new approach.

2.The explanations for Figure 2 and Figure 3 are not detailed enough, making the proposed method hard to understand.

3.It is suggested that the paper be targeted to a computer vision conference rather than a machine learning venue, given the limited theoretical support.

**Other Strengths And Weaknesses:**

Strengths:

1.The paper presents a comprehensive framework that effectively adapts visual in-context learning to OCR tasks, offering a practical solution for both text removal and segmentation.

2.It demonstrates thorough experimental validation across multiple benchmarks, which supports the claimed improvements of the proposed methods.

Weaknesses:

1. Overall, the work represents an incremental improvement in visual in-context learning for OCR tasks; both the context-aware aggregation and self-prompting strategies lack significant technical novelty and robust theoretical justification.

2. The paper proposes to recast the task demonstration by forming an explicit chain rather than a simple input-output pair, but it fails to discuss the impact of this design on training efficiency.

3. The method is tightly confined to OCR tasks, as it leverages the inherent logical relationship between text removal and segmentation—where the removed text is exactly the text to be segmented—limiting its extensibility to other domains.

**Questions For Authors:**

Please refer to the weaknesses section for the detailed questions, particularly regarding the limited scalability beyond OCR tasks.

**Relation To Broader Scientific Literature:**

The paper's contributions can be seen as an incremental improvement over existing V-ICL approaches like SegGPT and Painter, integrating task chaining, context-aware aggregation, and self-prompting strategies to better address OCR challenges.

**Theoretical Claims:**

The paper does not include any formal proofs, as its theoretical claims are supported primarily by intuitive reasoning and empirical evidence.

---

> ### Author Rebuttal · Authors · 2025-03-31
>
> ### W1 (Limited Contribution & Novelty)
> >...incremental improvement in visual in-context learning...lack significant technical novelty...
>
> We respectfully disagree with this and argue that our work, **rather than being merely incremental, offers substantial technical novelty and valuable contributions**:
>
> 1. *Novelty*: Beyond CAA and SP, we are **the first to explore visual task-chaining to enhance multi-task performance**, which is a novelty consistently acknowledged by other reviewers. Furthermore, as suggested by Reviewers qfVE & L63m, its **promising generalized scalability** has been experimentally validated across various tasks. Thus, we believe this simple yet effective design warrants recognition for its novelty and represents a significant contribution to visual-ICL.
> 2. *Contributions*: We'd like to emphasize that we propose **the first visual-ICL paradigm for OCR**, achieving SOTA performance against both generalist and specialist models. Notably, we find that our ConText **emerges with in-context understanding of visual-specific instructions in a training-free manner** (see **Figure 6**), e.g., removing and segmenting the text of the specified color, thereby facilitating flexible user-model interaction. We believe these valuable insights will encourage further developments in visual-ICL.
>
> ---
> ### W2 & S3 (Theoretical Support & Inappropriate Submission)
> >...lack robust theoretical justification...paper be targeted to a computer vision conference rather than a machine learning venue...
>
> Our method may lack theoretical analysis, but we argue it is grounded by **robust experimental findings and the underlying mechanisms of ICL** (please refer to Appendix A). We contend that this does not render it unsuitable for ICML, as **no stringent theoretical support is required for submissions.** This openness has fostered a series of impressive theoretical-free works in computer vision presented at ICML, such as the notable CLIP [1], the multi-modal generalist mPLUG-2 [2], and the OCR framework UPOCR [3]. Meanwhile, this embrace of diversity is also reflected in *Reviewer Instruction*: "*reviewers are encouraged to be open-minded in terms of potential strengths...particularly for application-driven ML papers.*" Therefore, while we understand your concerns, we respectfully hope for your inclusive understanding. We believe that our paper, which presents **a powerful and practical visual-ICL OCR paradigm with valuable insights and contributions** (as agreed upon by other reviewers), deserves consideration at this conference.
>
> ---
> ### W3 (Task-chaining Efficiency)
> >...it fails to discuss the impact of this design on training efficiency.
>
> Thanks for raising this. It is claimed that **a single task chain can address multiple tasks simultaneously, while the input-output baseline requires a separate training sample for each task**, e.g., one for segmentation and another for removal. As such, one sample of task-chaining effectively equals two samples from the baseline, which should yield **no significant training efficiency gap** between these approaches. As shown in the following table (HierText dataset), task-chaining adds merely **+0.6 minutes/epoch** compared to the baseline, indicating a quite acceptable level of training overhead. We will include this analysis.
> |Method|Training Time (per epoch)|
> |-|-|
> |SegGPT|3.2 min|
> |SegGPT+task-chaining|+0.6 min|
>
> ---
> ### W4 (Scalability)
> >...confined to OCR...limiting its extensibility to other domains.
>
> We agree that task-chaining utilizes inter-task logic for collective benefits, but we contend that **it is not merely limited to OCR domains**, and can serve as a flexible pipeline to **bridge different tasks with promising scalability:**
>
> 1. Intuitively, the segmentation-removal concept could be directly applied to the **natural image domain.** To verify this, we utilize one natural image removal benchmark, PIPE [4]. The results in the table below demonstrate ConText's superiority in the natural domain.
>
> |Method (PIPE dataset)|PSNR (Rem.)|fgIoU (Seg.)|
> |-|-|-|
> |SegGPT|31.72|58.92|
> |ConText|**35.33**|**62.76**|
>
> 2. Additionally, beyond the removal-segmentation logic, we argue that there are **inherent logical connections among various vision tasks that can be explored implicitly.** By scaling this task-in-chaining concept, we can enhance other image-level tasks, e.g., *watermark removal, edge detection, and denoising* (please refer to our responses to **W1-Reviewer L63m & qfVE**).
>
> ---
> ### S1 & 2 (Writing Issue)
> >...weakly written...Figure are not detailed...
>
> Thanks for this and we will make the suggested improvements.
>
> ---
> >### References
> >[1] Learning Transferable Visual Models From Natural Language Supervision, ICML'21
>
> >[2] mPLUG-2: A Modularized Multi-modal Foundation Model
> Across Text, Image and Video, ICML'23
>
> >[3] UPOCR: Towards Unified Pixel-Level OCR Interface, ICML'24
>
> >[4] Paint by Inpaint: Learning to Add Image Objects by Removing Them First, CVPR'25.

---

### Official Review · Reviewer_AZf8 · 2025-03-12

**Overall Recommendation:** 4

**Summary:**

The authors present a visual in-context learning (V-ICL) approach, ConText, for fine-grained text recognition tasks (segmentation and removal). In order to accomplish this they focus on three novelties:

1. Task-chaining: The two tasks are chained together instead of being done independently to leverage inter-task correspondence and enhance the generalised capability. The tasks would have to be related however for this benefit to manifest
2. Context aware aggregation (through attention mechanisms): helps leverage learnability from other image-label pairs in the given context, instead of simple addition
3. Self-prompting: helps select the right demonstrations; during training the most relevant demonstrations are selected dynamically which makes the more adaptive and helps deal with the context-homogeneity issue inherent in older models due to random or fixed demonstrations.

During training the standard masked-based strategy is used.

Overall, the authors show that their approach is superior to single-task methods across three datasets when considering both other generalists and specialists. Ablations are also presented to provide compelling qualitative evidence.

**Claims And Evidence:**

Yes, the authors show results supporting claims

**Essential References Not Discussed:**

No

**Experimental Designs Or Analyses:**

Yes, the experiment description in the ablation makes sense

**Methods And Evaluation Criteria:**

The datasets and metrics make sense for this task

**Other Comments Or Suggestions:**

Both the text and most diagrams feel very crammed with information, instead of fitting everything into the given page limit it would be better to be much more selective w.r.t the core information.

**Other Strengths And Weaknesses:**

Strengths
- Overall I think this is a novel paper that extends the current V-ICL space to leverage chain-of-thought and help produce good results for text-tasks without need for fine-tuning or human effort in curating labelling samples (through self-prompting)


Weaknesses
- May need to consider extra cost of self-prompting and context aggregation when evaluating this compared to competing methods

**Questions For Authors:**

- How does ConText perform on document-type datasets (i.e. dense text) such as DocLayNet
- I couldn't find an ablation on retrieval (nearest neighbour) vs self-prompting and wondering if former can be a lower cost way
- Would be interesting to understand in what particular scenarios (i.e. types of images) chaining is better than joint training (and in which it is not)

**Relation To Broader Scientific Literature:**

Text Removal/Segmentation is often a component of OCR pipelines, however these require explicit fine-tuning instead of in-prompt learning

(Visual) In-context Learning has leveraged prompt context to provide examples and avoid fine-tuning. However, currently tasks are done independently and not chained together (and also don't have the self-prompting and context-aggregation present in ConText)

Chain-of-though reasoning, popular with models such as R1, however hasn't been explored much for the visual modality

**Theoretical Claims:**

N/A

---

> ### Author Rebuttal · Authors · 2025-03-31
>
> ### W1 (Designed Modules Efficiency)
> >...extra cost of self-prompting and context aggregation...
>
> Below we report the additional costs of these modules, and find that
>
> 1. Self-prompting (SP) incurs an additional training burden of **+0.4 minutes/epoch**. However, this cost is deemed acceptable due to the moderate engagement of SP (SP-0.2) during training. Moreover, SP is not utilized during inference, yielding **no additional computational burden for inference.**
> 2. Context-aware aggregation (CAA) introduces extra computational costs during both training and inference. However, as a lightweight cross-attention module, it only **increases model complexity by a manageable 2%**. Consequently, it leads to a mere **increase of +0.03 seconds (+0.8 min/epoch) during inference (training).**
>
> Based on this, we can more safely conclude these modules indicate **a reasonable level of computational efficiency.** We will add this analysis.
> | Method|Training Time (per epoch) | Inference Time| Model FLOPs |
> |-|-|-|-|
> |Baseline|3.8 min|0.09 sec |666.76G|
> |Baseline + SP|+0.4 min|+0 sec|+0G (0%) |
> |Baseline + CAA | +0.8 min | +0.03 sec | +17.20G (2%) |
> |Baseline + CAA + SP| +1.0 min | +0.03 sec | +17.20G (2%) |
>
> **The results are evaluated on HierText, and FLOPs and inference time are calculated by forwarding one 512x512 image on one A100, with the inference time reported in seconds.*
>
> ---
> ### Q1 (Performance on Document Case)
> >How does ConText perform on document-type datasets (i.e. dense text) such as DocLayNet.
>
> We believe that **ConText could address this case due to its superior performance on HierText [1], which collects a substantial number of high-fidelity, real-world document-based samples** (please refer to [1] for some visualized examples and its data curation in *"Section 3"*). We will add more these visualized samples in the updated version. Since **DocLayNet lacks pixel-level stroke and removal annotations**, we may not be able to provide valid numerical evaluations during this rebuttal period. We hope for your understanding and plan to explore this further in the future.
>
> ---
> ### Q2 (SP *v.s.* Nearest Neighbor Retrieval)
> >...ablation on retrieval (nearest neighbor) vs self-prompting...
>
> Compared to SP, we agree with you that **the retrieval strategy should exhibit lower training costs due to its direct prompt-specific assignment.** To verify this, we adopt DINOv2 [2], a leading visual retriever [3-4], to assign the nearest counterpart for each sample as the in-context prompt during training. As shown in the table below, this once-for-all solution demonstrates slightly better training efficiency (**-0.35 minutes/epoch**) than SP. However, it is observed that **this retrieval-based selection yields inferior ICL ability**, as it has a narrower prompt selection scope compared to the adaptive SP. We will add this study.
>
> | Method| Training Time (per epoch) | Seg. (RS / GT) fgIoU↑|
> |-|-|-|
> |ConText + Retrieval|  **4.45 min** | 72.11 / 73.25  |
> |ConText + SP | 4.8 min | **74.86** / **78.12**    |
>
> **The above results were obtained using HierText. The training time is reported in minutes per epoch, and pre-processing time of retrieval is not calculated. RS (GT) refers to the randomly selected (ground-truth) demonstration case.*
>
> ---
> ### Q3 (Task-chaining *v.s.* Joint-training)
> >...what particular scenarios (i.e. types of images) chaining is better than joint training (and in which it is not)
>
> Thanks for raising this. Here we provide a corresponding case-by-case study as follows:
>
> 1. For those high-performance samples, as shown in Figure 4, we find that most of them contain **hard-to-recognize visual patterns**, e.g., texts with similar background colors (the last sample), small textual fonts (the 5th sample), and character-like objects (the 2nd and 3rd samples). In these cases, task-chaining explicitly highlights the prompted patterns across tasks, thereby yielding significant improvements.
> 2. Meanwhile, we also identify a small number of samples that exhibit limited or even worse performance compared to joint-training. Most of them have relatively **low resolution and noisy annotations in text regions**. In this way, simply relying on task-chaining may exacerbate the accumulation of these noises, leading to performance results that fall below expectations compared to joint-training.
>
> In conclusion, **task-chaining shows overall superiority over joint-training.** We will add this discussion.
>
> ---
> ### S1 (Crammed formatting)
> >...it would be better to be much more selective w.r.t the core information.
>
> Thanks for this advice and we will polish our formatting for better clarity.
>
> ---
> >### References
> >[1] Towards End-to-End Unified Scene Text Detection and Layout Analysis, ICDAR'23.
>
> >[2] DINOv2: Learning Robust Visual Features without Supervision, TMLR'24.
>
> >[3] Retrieval-augmented embodied agents, CVPR'24.
>
> >[4] FORB: a flat object retrieval benchmark for universal image embedding, NeurIPS'23.

---

### Official Review · Reviewer_L63m · 2025-03-12

**Overall Recommendation:** 4

**Summary:**

This paper explores the application of visual in-context learning to fine-grained text recognition tasks, including text segmentation and removal. Rather than employing single-task solutions, the authors propose a task-chain prompt framework that connects multiple tasks. Through extensive experimentation, they demonstrate state-of-the-art results on both text removal and text segmentation benchmarks while showing the effectiveness of the task-chaining approach compared to single-task methods.

**Claims And Evidence:**

yes

**Essential References Not Discussed:**

None

**Experimental Designs Or Analyses:**

Please refer to the weaknesses

**Methods And Evaluation Criteria:**

The evaluations make sense to me.

**Other Comments Or Suggestions:**

NA

**Other Strengths And Weaknesses:**

- **Strengths**
  - The paper is well-written, with good presentation and extensive experimental evaluation.
  - The work successfully extends visual in-context learning from single-task scenarios to task-chaining prompts, enhancing the model's ICL capabilities through multi-task reasoning.
  - Extensive benchmarking demonstrates that the approach achieves state-of-the-art performance on both text removal and segmentation tasks, with relatively huge improvements over other ICL methods.

- **Weaknesses**
  - The idea of extending single-task in-context learning to multi-task is interesting, is it possible to generalize to other tasks? Also, the task of text segmentation and removal only has two tasks, how would the framework perform with three or more chained tasks? A discussion of the generalizability of the framework would be helpful.
  - Table 5 shows that self-prompting has limited effect in randomly selected settings. Is there any interpretation of this? Additionally, the threshold selection for self-prompting lacks comprehensive evaluation - including results across a range of thresholds (e.g., SP = [0, 0.1, 0.2, ... 1.0]) would better demonstrate how to optimize this parameter.
  - The ablation study is conducted only on SegGPT rather than on the proposed ConText model.
  - While the paper demonstrates significant improvements from ConText, it lacks direct comparisons between individual task paths (ConText → Rem and ConText → Seg) against both other methods and the combined approach (ConText → Rem + Seg). While there are comparisons shown on the SegGPT and Painter model, a direct comparison of the proposed model would give more intuitive insights of the advantage of task-in-chain over a single task.

**Questions For Authors:**

Please see the weaknesses.

**Relation To Broader Scientific Literature:**

The paper explores how to extend single-task in-context learning to multi-task in-context learning, using text removal and text segmentation as a case study.

**Theoretical Claims:**

correct

---

> ### Author Rebuttal · Authors · 2025-03-31
>
> ### W1 (Beyond two tasks)
> >...is it possible to generalize to other tasks?...how would the framework perform with three or more chained tasks...
>
> We'd like to argue that **our method can intuitively be integrated with other tasks (even image-level tasks) alongside chaining extension**. Except for adapting ConText to watermark removal (please refer to **W1-Reviewer qfVE**), we have conducted two tasks: image *edge detection* and *denoising*. As shown in the table below (HierText), our method also shows promising performance on these new tasks, and the progressive task-chaining could lead to a global improvement. Overall, these results help demonstrate the **task-wise scalability of ConText**. We will add this analysis.
> |Method|MAE↓ (Edge.)|PSNR↑ (Den.)| PSNR↑ (Rem.) | fgIoU↑ (Seg.)|
> |-|-|-|-|-|
> | Painter -> Edge.|29.78|-|-|-|
> | ConText -> Edge.|**22.74**|-|-|-|
> | Painter -> Den.|-|30.45|-|-|
> | ConText -> Den.|-|**34.57**|-|-|
> | Painter -> Edge. + Den.  |27.17|32.68|-|-|
> | ConText -> Edge. + Den. |**19.79**|**36.83**|-|-|
> | Painter -> Edge. + Rem. + Seg.  |26.48|-|30.12|66.43|
> | ConText -> Edge. + Rem. + Seg.  |**17.76**|-|**40.23**|**76.61**|
> | Painter -> Edge. + Den. + Rem. + Seg.|25.22|34.73|31.19|68.65|
> | ConText -> Edge. + Den. + Rem. + Seg.|**16.14**|**38.19**|**41.26**|**77.98**|
>
> *For edge detection (Edge.), we use the Canny operator for the whole image regions. For denoising (Den.), we follow [1] and add the image with noising. To chain these tasks, we keep the noised counterpart as the 1st input, set the original image as the 2nd one, and the edge-detected image, along with the removal and segmentation masks, as the following parts.*
>
> ---
> ### W2 (Discussion on Self-prompting Strategy)
> >...self-prompting has limited effect in randomly selected settings...(e.g., SP = [0, 0.1, 0.2, ... 1.0]) would better demonstrate how to optimize this parameter.
>
> The observed weakened performance could essentially be attributed to the **in-context empowerment mechanism** of the self-prompting (SP). SP enables the model to maintain in-context generalization by guiding it to perform shortcut-like reasoning based on the self-demonstrations. Compared to the no-SP process, where no valid counterparts are provided during the training, **the model with SP may reduce the complexity of task reasoning by learning valid prompted patterns from the given self-demonstrations**. Consequently, this simplification shall reasonably yield a performance drop in demonstration-free (randomly selected) cases.
>
> The table below provides a more detailed assessment of SP. **As the involvement of SP increases, the model generally exhibits decreased demonstrate-free reasoning ability and improved in-context learnability,** highlighting the significance of proper utilization of SP. We will add this discussion.
> | Method   | Seg. (RS / GT) fgIoU   | Rem. (RS / GT) PSNR  |
> | -------- | -------------- | -------------- |
> | Baseline | 79.14 / +0.65  | 38.59 / +0.37  |
> | SP-0.1   | 78.45 / +2.12  | 38.02 / +0.77   |
> | SP-0.2   | 78.02 / +3.98  | 37.67 / +0.92  |
> | SP-0.3   | 77.94 / +4.05  | 37.12 / +1.24  |
> | SP-0.4   | 77.63 / +4.17  | 36.82 / +1.45  |
> | SP-0.5   | 77.39 / +5.04  | 36.52 / +1.94      |
> | SP-0.6   | 77.14 / +5.83  | 36.12 / +2.13      |
> | SP-0.7   | 76.64 / +6.44  | 35.82 / +2.75      |
> | SP-0.8   | 76.33 / +6.73      | 35.37 / +3.47      |
> | SP-0.9   | 76.12 / +6.95      | 35.03 / +3.96      |
> | SP-1.0   | 75.94 / +7.04      | 34.81 / +4.37      |
>
> **The table follows same setting of Table 5. RS (GT) refers to the randomly selected (ground-truth) demonstration case. Here Baseline refers to the baseline method + CAA.*
>
> ---
> ### W3 & 4 (Ablation Studies on ConText)
> >The ablation study is conducted only on SegGPT rather than on the proposed ConText model.
>
> Due to the page limit, we have presented several ablations on ConText at **Appendix B.3 & B.6**, including the **influence of synthetic training datasets**, **masking ratio**, and **different inference mechanisms**. We will adjust the formatting for better clarity.
>
> >...it lacks direct comparisons between individual task paths...
>
> Thanks for this advice. Below we present the suggested single-task-tuned experiments on HierText. Clearly, our single-task-tuned ConText outperforms multi-task baselines, and our task-chaining ConText further shows a significant improvement over its single-task-oriented counterparts, validating the advantage of the task-in-chain concept (our response to your *W1* could also help validate this). We will add this analysis.
> |Method|PSNR (Rem.)| fgIoU (Seg.)|
> |-|-|-|
> | Painter -> Rem. + Seg.|28.17|60.60|
> | SegGPT -> Rem. + Seg.  |28.16|65.23|
> | ConText -> Rem.|**37.35**|-|
> | ConText -> Seg.|-|**70.88**|
> | ConText -> Rem. + Seg.|**39.48**|**74.86**|
>
> **We implement single-task-tuned ConText through implementing the designed modules by only fusing the single task feature.*
>
> ---
> >### References
> >[1] Visual Prompting via Image Inpainting, NeurIPS'22.

---

> > ### Comment · Reviewer_L63m · 2025-04-03
> >
> > Thanks for the detailed response and the additional results. My questions have been resolved, and I have increased my score to 4. It would be helpful to also include a discussion on what types of chaining tasks could benefit from the proposed design along with the additional experiment results, which is also mentioned by reviewer AZf8.

---

> > > ### Author Response · Authors · 2025-04-04
> > >
> > > Thank you for your feedback. We are pleased to hear that our rebuttal has addressed your concerns. We also agree that discussing various types of chaining tasks is valuable. To this end, we‘d like to provide a case-by-case analysis as follows:
> > >
> > > 1. *Highly Relevant Tasks*: For tasks that are highly related, such as text/watermark/object removal and segmentation, task-wise chaining can lead to significant collective improvements across all sub-tasks (+3.98 mIoU and +2.13 PSNR for OCR tasks). This is because these tasks are explicitly logically chained (i.e., the removed object directly corresponds to the segmented target), and our task chaining effectively highlights these patterns, facilitating the model's multi-task learning.
> > >
> > > 2. *Implicitly Relevant Tasks*: Additionally, we implement two low-level tasks with comparably implicit logical patterns:edge detection and image denoising. We find that these tasks also contribute to a certain degree of collective improvement. This underscores the potential correlation among various vision tasks. However, since these task-level chains do not demonstrate explicit relationships like removal-segmentation, the observed collective improvement is less highlighted than that of the more explicitly chained tasks.
> > >
> > > 3. *Bridging All*: Finally, we explore the potential of bridging all these tasks regardless of explicit logical connections and find that this approach can yield further gains. Moreover, it is revealed that denoising, which precedes segmentation and removal, can enhance the overall model's performance by relieving some original low-performing cases with noisy annotations. Meanwhile, segmentation and removal tasks could be improved from edge detection, as it involves more fine-grained texture patterns.
> > >
> > > Overall, the above analysis illustrates the benefits of task-chaining and emphasizes the importance of understanding task-wise relationships. Combing this analysis with our response of Q2-Reviewer AZf8 shall provide a more comprehensive discussion on task-chaining. We will add this at the updated version.
> > >
> > > Thanks again for your efforts and increasing the score of our work.

---

### Official Review · Reviewer_qfVE · 2025-03-20

**Overall Recommendation:** 4

**Summary:**

This paper proposes ConText, an adaptation of the visual in-context learning (V-ICL) paradigm specifically tailored for optical character recognition tasks, focusing on text removal and segmentation. To address the single-step reasoning bottleneck in existing V-ICL methods, ConText introduces a task-chaining compositor, sequentially linking text removal and segmentation tasks. Additionally, the authors propose a context-aware aggregation module to enhance latent query representation and introduce self-prompting to maintain robust in-context reasoning and prevent overly specialized, context-free inference. Extensive experiments on multiple benchmarks demonstrate that ConText significantly outperforms existing V-ICL generalist and specialist methods, achieving new state-of-the-art results in both text removal and segmentation tasks.

**Claims And Evidence:**

The paper makes three primary claims:

First, the authors claim that existing methods limit models' in-context learning (ICL) capability by restricting prompts to image-label pairs, and propose a task-chaining compositor to enhance in-context reasoning. Second, they argue that their proposed context aggregation module effectively improves contextual understanding. Third, the authors assert that the inherent heterogeneity of text is more complex than object-level scenes, motivating their simple yet effective self-prompting strategy.

Collectively, these claims underpin the development of ConText, which achieves state-of-the-art performance on several benchmarks for text removal and segmentation, verifying the combined effectiveness of the proposed methods. Individually, the authors provide supporting evidence for each claim; for instance, Fig. 5 demonstrates that the task-chaining compositor significantly enhances out-of-domain generalization capabilities.

**Essential References Not Discussed:**

No

**Experimental Designs Or Analyses:**

The experimental design is sound, and the analysis is comprehensive.

**Methods And Evaluation Criteria:**

The proposed methodology is clearly written and logically aligns with the ICL framework for text removal and segmentation. The benchmarks, baselines, and evaluations are comprehensive.

**Other Comments Or Suggestions:**

N/A

**Other Strengths And Weaknesses:**

Strengths:

The paper introduces fresh ideas in visual in-context learning, particularly through the task-chaining compositor and context aggregation module, which enhance task chaining. The methodology is easy to understand yet effective, offering potential insights for future research in visual in-context learning.

Weaknesses:

While the task-chaining compositor is shown to be effective, the paper only explores chaining two tasks, i.e. text removal and text segmentation. This may be due to dataset limitations, but extending the approach to additional tasks, such as watermark removal or text registration, would be valuable for further exploration.

**Questions For Authors:**

N/A

**Relation To Broader Scientific Literature:**

N/A

**Theoretical Claims:**

No theoretical claims were evaluated.

---

> ### Author Rebuttal · Authors · 2025-03-31
>
> ### W1 (Additional Task Exploration)
> >While the task-chaining compositor is shown to be effective...This may be due to dataset limitations, but extending the approach to additional tasks, such as watermark removal or text registration, would be valuable for further exploration.
>
> Thanks for this valuable advice. We also appreciate your understanding regarding the challenging scarcity of relevant OCR benchmarks. Considering data availability, we have additionally explored our ConText on 1 prevailing *watermark removal* benchmark, CLWD [1]. The table below reports the results of SegGPT and a leading specialist [2]. Clearly, our approach demonstrates superior performance, further validating the **task-wise generalizability of ConText** (We also kindly invite you to refer to our response for **W1-Reviewer L63m**, where we provide additional generalizations to more tasks).  We will include this analysis.
>
> | Method | PSNR↑ (Rem.) | fgIoU↑ (Seg.) |
> | ------- | -- | ----|
> | SegGPT (generalist) |   30.11     |  74.42     |
> | PFMNet [2] (specialist) |   39.45       | 79.09  |
> | ConText |   **40.73**      |  **82.16**     |
>
> **CLWD includes both pixel-level binary watermark masks and the removal images. We utilize its training set to train SegGPT and ConText with similar learning strategy. Rem. (Seg.) refers to the removal (segmentation) task.*
>
> ---
> >### Reference
> >[1] Wdnet: Watermarkdecomposition network for visible watermark removal, CVPR22.
>
> >[2] Fine-grained Visible Watermark Removal, ICCV23.

---

### Decision · Program_Chairs · 2025-05-01

**Decision:**

Accept (poster)

**Comment:**

This paper presents a visual in-context learning approach for fine-grained text recognition tasks, including segmentation and removal.

After the rebuttal, three reviewers (qfVE, L63m, and AZf8) provided positive comments, recognizing the paper's strong performance and novel ideas. The other reviewer (qCFq) did not respond to the authors' feedback.

The Area Chair (AC) agrees with most of the reviewers and recommends accepting the paper. Additionally, it is strongly suggested that the authors incorporate the content discussed in the feedback to make the paper more comprehensive.